# HINDSIGHT TRUST REGION POLICY OPTIMIZATION

## ABSTRACT

As reinforcement learning continues to drive machine intelligence beyond its conventional boundary, unsubstantial practices in sparse reward environment severely limit further applications in a broader range of advanced fields. Motivated by the demand for an effective deep reinforcement learning algorithm that accommodates sparse reward environment, this paper presents Hindsight Trust Region Policy Optimization (HTRPO), a method that efficiently utilizes interactions in sparse reward conditions to optimize policies within trust region and, in the meantime, maintains learning stability. Firstly, we theoretically adapt the TRPO objective function, in the form of the expected return of the policy, to the distribution of hindsight data generated from the alternative goals. Then, we apply Monte Carlo with importance sampling to estimate KL-divergence between two policies, taking the hindsight data as input. Under the condition that the distributions are sufficiently close, the KL-divergence is approximated by another $f$-divergence. Such approximation results in the decrease of variance and alleviates the instability during policy update. Experimental results on both discrete and continuous benchmark tasks demonstrate that HTRPO converges significantly faster than previous policy gradient methods. It achieves effective performances and high data-efficiency for training policies in sparse reward environments.

## 1 INTRODUCTION

Reinforcement Learning has been a heuristic approach confronting a great many real-world problems from playing complex strategic games (Mnih et al., 2015; Silver et al., 2016; Justesen et al., 2019) to the precise control of robots(Levine et al., 2016; Mahler & Goldberg, 2017; Quillen et al., 2018), in which policy gradient methods play very important roles(Sutton et al., 2000; Deisenroth et al., 2013). Among them, the ones based on trust region including Trust Region Policy Optimization (Schulman et al., 2015a) and Proximal Policy Optimization (Schulman et al., 2017) have achieved stable and effective performances on several benchmark tasks. Later on, they have been verified in a variety of applications including skill learning(Nagabandi et al., 2018), multi-agent control(Gupta et al., 2017), imitation learning(Ho et al., 2016), and have been investigated further to be combined with more advanced techniques(Nachum et al., 2017; Houthooft et al., 2016; Heess et al., 2017).

One unresolved core issue in reinforcement learning is efficiently training the agent in sparse reward environments, in which the agent is given a distinctively high feedback only upon reaching the desired final goal state. On one hand, generalizing reinforcement learning methods to sparse reward scenarios obviates designing delicate reward mechanism, which is known as reward shaping(Ng et al., 1999); on the other hand, receiving rewards only when precisely reaching the final goal states also guarantees that the agent can focus on the intended task itself without any deviation.

Despite the extensive use of policy gradient methods, they tend to be vulnerable when dealing with sparse reward scenarios. Admittedly, policy gradient may work in simple and sufficiently rewarding environments through massive random exploration. However, since it relies heavily on the expected return, the chances in complex and sparsely rewarding scenarios become rather slim, which often makes it unfeasible to converge to a policy by exploring randomly.

Recently, several works have been devoted to solving the problem of sparse reward, mainly applying either hierarchical reinforcement learning (Kulkarni et al., 2016; Vezhnevets et al., 2017; Le et al., 2018; Marino et al., 2019) or a hindsight methodology, including Hindsight Experience Replay

(Andrychowicz et al., 2017), Hindsight Policy Gradient (Rauber et al., 2019) and their extensions (Fang et al., 2019; Levy et al., 2019). The idea of Hindsight Experience Replay(HER) is to regard the ending states obtained through the interaction under current policy as alternative goals, and therefore generate more effective training data comparing to that with only real goals. Such augmentation overcomes the defects of random exploration and allows the agent to progressively move towards intended goals. It is proven to be promising when dealing with sparse reward reinforcement learning problems.

For Hindsight Policy Gradient(HPG), it introduces hindsight to policy gradient approach and improves sample efficiency in sparse reward environments. Yet, its learning curve for policy update still oscillates considerably. Because it inherits the intrinsic high variance of policy gradient methods which has been widely studied in Schulman et al. (2015b), Gu et al. (2016) and Wu et al. (2018). Furthermore, introducing hindsight to policy gradient methods would lead to greater variance (Rauber et al., 2019). Consequently, such exacerbation would cause obstructive instability during the optimization process.

To design an advanced and efficient on-policy reinforcement learning algorithm with hindsight experience, the main problem is the contradiction between on-policy data needed by the training process and the severely off-policy hindsight experience we can get. Moreover, for TRPO, one of the most significant property is the approximated monotonic converging process. Therefore, how these advantages can be preserved when the agent is trained with hindsight data also remains unsolved.

In this paper, we propose a methodology called Hindsight Trust Region Policy Optimization (HTRPO). Starting from TRPO, a hindsight form of policy optimization problem within trust region is theoretically derived, which can be approximately solved with the Monte Carlo estimator using severely off-policy hindsight experience data. HTRPO extends the effective and monotonically iterative policy optimization procedure within trust region to accommodate sparse reward environments. In HTRPO, both the objective function and the expectation of KL divergence between policies are estimated using generated hindsight data instead of on-policy data. To overcome the high variance and instability in KL divergence estimation, another $f$-divergence is applied to approximate KL divergence, and both theoretically and practically, it is proved to be more efficient and stable.

We demonstrate that on several benchmark tasks, HTRPO can significantly improve the performance and sample efficiency in sparse reward scenarios while maintains the learning stability. From the experiments, we illustrate that HTRPO can be neatly applied to not only simple discrete tasks but continuous environments as well. Besides, it is verified that HTRPO can be generalized to different hyperparameter settings with little impact on performance level.

## 2 PRELIMINARIES

**Reinforcement Learning Formulation and Notation.** Consider the standard infinite-horizon reinforcement learning formulation which can be defined by tuple $(\mathcal{S}, \mathcal{A}, \pi, \rho_0, r, \gamma)$. $\mathcal{S}$ represents the set of states and $\mathcal{A}$ denotes the set of actions. $\pi : \mathcal{S} \to \mathcal{P}(\mathcal{A})$ is a policy that represents an agent's behavior by mapping states to a probability distribution over actions. $\rho_0$ denotes the distribution of the initial state $s_0$. Reward function $r : \mathcal{S} \to \mathbb{R}$ defines the reward obtained from the environment and $\gamma \in (0, 1)$ is a discount factor. In this paper, the policy is a differentiable function regarding parameter $\theta$. We follow the standard formalism of state-action value function $Q(s, a)$, state value function $V(s)$ and advantage function $A(s, a)$ in Sutton & Barto (2018). We also adopt the definition of $\gamma$-discounted state visitation distribution as $\rho_\theta(s) = (1 - \gamma) \sum_{t=0}^{\infty} \gamma^t P(s_t = s)$ (Ho et al., 2016), in which the coefficient $1 - \gamma$ is added to keep the integration of $\rho_\theta(s)$ as 1. Correspondingly, $\gamma$-discounted state-action visitation distribution (Ho et al., 2016), also known as occupancy measure (Ho & Ermon, 2016), is defined as $\rho_\theta(s, a) = \rho_\theta(s) \times \pi_\theta(a|s)$, in which $\pi_\theta(a|s)$ stands for the policy under parameter $\theta$.

**Trust Region Policy Optimization(TRPO).** Schulman et al. (2015a) proposes an iterative trust region method that effectively optimizes policy by maximizing the per-iteration policy improvement. The optimization problem proposed in TRPO can be formalized as follows:

$$\max_\theta \ L_{TRPO}(\theta) \tag{1}$$

$$s.t. \quad \mathbb{E}_{s \sim \rho_{\tilde{\theta}}(s)} \left[ D_{KL}(\pi_{\tilde{\theta}}(a|s) || \pi_{\theta}(a|s)) \right] \leq \epsilon \tag{2}$$

in which $\rho_{\tilde{\theta}}(s) = \sum_{t=0}^{\infty} \gamma^t P(s_t = s)$. $\theta$ denotes the parameter of the new policy while $\tilde{\theta}$ is that of the old one. Trajectory is represented by $\tau = s_1, a_1, s_2, a_2, ....$. The objective function $L_{TRPO}(\theta)$ can be given out in the form of expeted return:

$$L_{TRPO}(\theta) = \mathbb{E}_{s,a \sim \rho_{\tilde{\theta}}(s,a)} \left[ \frac{\pi_{\theta}(a|s)}{\pi_{\tilde{\theta}}(a|s)} A_{\tilde{\theta}}(s,a) \right] \tag{3}$$

**Hindsight Policy Gradient(HPG).** After generalizing the concept of hindsight, Rauber et al. (2019) combines the idea with policy gradient methods. Though goal-conditioned reinforcement learning has been explored for a long time and actively investigated in recent works(Peters & Schaal, 2008; Schaul et al., 2015; Andrychowicz et al., 2017; Nachum et al., 2018; Held et al., 2018; Nair et al., 2018; Veeriah et al., 2018), HPG firstly extends the idea of hindsight to goal-conditioned policy gradient and shows that the policy gradient can be computed in expectation over all goals. The goal-conditioned policy gradient is derived as follows:

$$\nabla_{\theta} \eta(\theta) = \mathbb{E}_g \left[ \mathbb{E}_{\tau \sim p_{\theta}(\tau|g)} \left[ \sum_{t=1}^{T-1} \nabla_{\theta} \log \pi_{\theta}(a_t \mid s_t, g) A_{\theta}(s_t, a_t, g) \right] \right] \tag{4}$$

Then, by applying hindsight formulation, it rewrites goal-conditioned policy gradient with trajectories conditioned on some other goal $g'$ using importance sampling (Bishop, 2016) to improve sample efficiency in sparse-reward scenarios.

In this paper, we propose an approach that introduces the idea of hindsight to TRPO, called **Hindsight Trust Region Policy Optimization(HTRPO)**, aiming to further improve policy performance and sample efficiency for reinforcement learning with sparse rewards. In Section 3 and Section 4, we demonstrate how to redesign the objective function and the constraints starting from TRPO respectively.

## 3 EXPECTED RETURN AND POLICY GRADIENTS OF HTRPO

In order to apply hindsight methodology, this section presents the main steps for the derivation of HTRPO objective function. Starting from the original optimization problem in TRPO, the objective function can be written in the following variant form:

$$L_{\tilde{\theta}}(\theta) = \mathbb{E}_{\tau \sim p_{\tilde{\theta}}(\tau)} \left[ \sum_{t=0}^{\infty} \gamma^t \frac{\pi_{\theta}(a_t|s_t)}{\pi_{\tilde{\theta}}(a_t|s_t)} A_{\tilde{\theta}}(s_t, a_t) \right] \tag{5}$$

The derivation process of this variant form is shown explicitly in Appendix A.1 and in Schulman et al. (2015a).

Given the expression above, we consider the goal-conditioned objective function of TRPO as a premise for hindsight formulation. Similar to equation 4, $L_{\tilde{\theta}}(\theta)$ can be correspondingly given out in the following form:

$$L_{\tilde{\theta}}(\theta) = \mathbb{E}_g \left[ \mathbb{E}_{\tau \sim p_{\tilde{\theta}}(\tau|g)} \left[ \sum_{t=0}^{\infty} \gamma^t \frac{\pi_{\theta}(a_t|s_t, g)}{\pi_{\tilde{\theta}}(a_t|s_t, g)} A_{\tilde{\theta}}(s_t, a_t, g) \right] \right] \tag{6}$$

For the record, though it seems that equation 6 makes it possible for off-policy learning, it can be used as the objective only when policy $\pi_{\theta}$ is close to the old policy $\pi_{\tilde{\theta}}$, i.e. within the trust region. Using severely off-policy data like hindsight experience will make the learning process diverge. Therefore, importance sampling need to be integrated to correct the difference of the trajectory distribution caused by changing the goal. Based on the goal-conditioned form of the objective function, the following theorem gives out the hindsight objective function conditioned on some goal $g'$ with the distribution correction derived from importance sampling.

**Theorem 3.1** (HTRPO Objective Function). For the original goal $g$ and an alternative goal $g'$, the object function of HTRPO $L_{\tilde{\theta}}(\theta)$ is given by:

$$L_{\tilde{\theta}}(\theta) = \mathbb{E}_{g'} \left[ \mathbb{E}_{\tau \sim p_{\theta}(\tau|g)} \left[ \sum_{t=0}^{\infty} \prod_{k=1}^{t} \frac{\pi_{\tilde{\theta}}(a_k|s_k, g')}{\pi_{\tilde{\theta}}(a_k|s_k, g)} \gamma^t \frac{\pi_{\theta}(a_t|s_t, g')}{\pi_{\tilde{\theta}}(a_t|s_t, g')} A_{\tilde{\theta}}(s_t, a_t, g') \right] \right], \tag{7}$$

in which, $\tau = s_1, a_1, s_2, a_2, ..., s_t, a_t$.

Appendix A.2 presents an explicit proof on how the hindsight-form objective function derives from equation 6. It will be solved under a KL divergence expectation constraint, which will be discussed in detail in Section 4. Intuitively, equation 7 provides a way to compute the expected return in terms of the advantage with new-goal-conditioned hindsight experiences which are generated from interactions directed by old goals.

Naturally, Theorem 3.2 gives out the gradient of HTRPO objective function that will be applied to solve the optimization problem. Detailed steps of computing the gradient is presented in Appendix A.3.

**Theorem 3.2** (Gradient of HTRPO Objective Function). For the original goal $g$ and an alternative goal $g'$, the gradient $\nabla_\theta L_{\tilde{\theta}}(\theta)$ of HTRPO object function with respect to $\theta$ is given by the following expression:

$$\nabla_\theta L_{\tilde{\theta}}(\theta) = \mathop{\mathbb{E}}_{g'} \left[ \mathop{\mathbb{E}}_{\tau \sim p_\theta(\tau|g)} \left[ \sum_{t=0}^{\infty} \prod_{k=1}^{t} \frac{\pi_{\tilde{\theta}}(a_k|s_k, g')}{\pi_{\tilde{\theta}}(a_k|s_k, g)} \gamma^t \frac{\nabla_\theta \pi_\theta(a_t|s_t, g')}{\pi_{\tilde{\theta}}(a_t|s_t, g')} A_{\tilde{\theta}}(s_t, a_t, g') \right] \right], \quad (8)$$

in which $\tau = s_1, a_1, s_2, a_2, ..., s_t, a_t$.

## 4 EXPECTATION OF KL DIVERGENCE ESTIMATION

This section firstly demonstrates some techniques, with strict proof, that can be used to estimate the expectation of KL-divergence and further reduce the variance, and then presents how hindsight is applied to the constraint function of TRPO.

In TRPO, the KL divergence expectation under $\rho_{\tilde{\theta}}(s)$ is estimated by averaging all the values of KL divergence. When they are respectively conditioned on all states collected using the old policy, this kind of estimation is exactly Monte Carlo estimation which is unbiased. However, when we only have access to hindsight experience data, the state distribution may inevitably change and the previous method for estimating the expectation of KL divergence is no longer valid. To solve this problem, we firstly transform the KL divergence to an expectation under occupancy measure $\rho_{\tilde{\theta}}(s, a) = \rho_{\tilde{\theta}}(s) \times \pi_{\tilde{\theta}}(a|s)$. It can be estimated using collected state-action pair $(s, a)$, whose changed distribution can be corrected by importance sampling. Then, by making use of another $f$-divergence, the variance of estimation is theoretically proved to be reduced so as to facilitating a more stable training.

The constraint function in KL-divergence can be naturally converted to a logarithmic form. Appendix B.1 provides a more explicit version of this conversion.

**Theorem 4.1** (Logarithmic Form of Constraint Function). Given two policies $\pi_{\tilde{\theta}}(a|s)$ and $\pi_\theta(a|s)$, the expectation of their KL-divergence over states $s \sim \rho_{\tilde{\theta}}(s)$ is written as:

$$\mathop{\mathbb{E}}_{s \sim \rho_{\tilde{\theta}}(s)} \left[ D_{KL}(\pi_{\tilde{\theta}}(a|s)||\pi_\theta(a|s)) \right] = \mathop{\mathbb{E}}_{s,a \sim \rho_{\tilde{\theta}}(s,a)} \left[ \log \pi_{\tilde{\theta}}(a|s) - \log \pi_\theta(a|s) \right] \quad (9)$$

However, simply expanding the KL-divergence into logarithmic form still leaves several problems unhandled. Firstly, the variance remains excessively high, which would cause considerable instability during the learning process. Secondly, current estimation of KL-divergence is of possible negativity. If encountering negative expectation of KL-divergence, the learning process would result in fatal instability.

The following Theorem 4.2 describes a technique to reduce the variance and Theorem 4.3 gives out the strict proof for the decrease of variance.

**Theorem 4.2** (Approximation of Constraint Function). For policy $\pi_{\tilde{\theta}}(a|s)$ and $\pi_\theta(a|s)$, and for $\eta = \pi_\theta(a|s) - \pi_{\tilde{\theta}}(a|s)$,

$$\mathop{\mathbb{E}}_{s,a \sim \rho_{\tilde{\theta}}(s,a)} \left[ \log \pi_{\tilde{\theta}}(a|s) - \log \pi_\theta(a|s) \right] = \mathop{\mathbb{E}}_{s,a \sim \rho_{\tilde{\theta}}(s,a)} \left[ \frac{1}{2} (\log \pi_{\tilde{\theta}}(a|s) - \log \pi_\theta(a|s))^2 \right]$$
$$+ \mathop{\mathbb{E}}_{s,a \sim \rho_{\tilde{\theta}}(s,a)} \left[ o(\eta^3) \right]. \quad (10)$$

Theorem 4.2 demonstrates that when $\theta$ and $\tilde{\theta}$ is of limited difference, the expectation of $\log \pi_{\tilde{\theta}}(a|s) - \log \pi_\theta(a|s)$ can be sufficiently estimated by the expectation of its square. The proof is provided in Appendix B.2. In fact, $\mathbb{E}_{s,a\sim\rho_{\tilde{\theta}}(s,a)} \left[ \frac{1}{2}(\log \pi_{\tilde{\theta}}(a|s) - \log \pi_\theta(a|s))^2 \right]$ is the expectation of an $f$-divergence, where $f(x) = \frac{1}{2}x(\log x)^2$. Noticeably, $f(x)$ is a strictly convex function when $x \in (\frac{1}{e}, \infty)$, and $f(1) = 0$.

Moreover, it is noteworthy that there are two corresponding major improvements through this kind of estimation. Firstly, it is guaranteed to reduce the variance which leads to a more stable performance. This merit will be explained in detail in Theorem 4.3. Another significant improvement is manifested in the elimination of negative KL-divergence, since the estimation presents itself in the form of a square which is always non-negative.

**Theorem 4.3** (Variance of Constraint Function). For policy $\pi_{\tilde{\theta}}(a|s)$ and $\pi_\theta(a|s)$, let Var denotes the variance of a variable. For any action $a \in \mathcal{A}$ and any state $s \in \mathcal{S}$, when $\log \pi_{\tilde{\theta}}(a|s) - \log \pi_\theta(a|s) \in [-0.5, 0.5]$, then

$$\operatorname*{Var}_{s,a\sim\rho_{\tilde{\theta}}(s,a)} \left[ \frac{(\log \pi_{\tilde{\theta}}(a|s) - \log \pi_\theta(a|s))^2}{2} \right] \leq \operatorname*{Var}_{s,a\sim\rho_{\tilde{\theta}}(s,a)} \left[ \log \pi_{\tilde{\theta}}(a|s) - \log \pi_\theta(a|s) \right]. \quad (11)$$

Theorem 4.3 illustrates that there is a decrease from the variance of $\log \pi_{\tilde{\theta}}(a|s) - \log \pi_\theta(a|s)$ to the variance of its square, and furthermore indicates that the variance is effectively reduced. The proof is given in detail in Appendix B.3. In fact, the closer it is between $\tilde{\theta}$ and $\theta$, the more the variance decreases.

Based on Theorem 4.1 to Theorem 4.3, in this paper, we adopt the following form of constraint condition:

$$\mathbb{E}_{s,a\sim\rho_{\tilde{\theta}}(s,a)} \left[ \frac{1}{2}(\log \pi_{\tilde{\theta}}(a|s) - \log \pi_\theta(a|s))^2 \right] \leq \epsilon. \quad (12)$$

In Theorem 4.4, we demonstrate that hindsight can also be introduced to the constraint function. The proof follows the methodology similar to that in Section 3, and is deducted explicitly in Appendix B.4.

**Theorem 4.4** (HTRPO Constraint Function). For the original goal $g$ and an alternative goal $g'$, the constraint between policy $\pi_{\tilde{\theta}}(a|s)$ and policy $\pi_\theta(a|s)$ is given by:

$$\mathbb{E}_{g'} \left[ \mathbb{E}_{\tau\sim p_\theta(\tau|g)} \left[ \frac{1}{2} \sum_{t=0}^{\infty} \prod_{k=1}^{t} \frac{\pi_{\tilde{\theta}}(a_k|s_k, g')}{\pi_{\tilde{\theta}}(a_k|s_k, g)} \gamma^t (\log \pi_{\tilde{\theta}}(a_t|s_t, g') - \log \pi_\theta(a_t|s_t, g'))^2 \right] \right] \leq \epsilon'. \quad (13)$$

in which $\epsilon' = \frac{\epsilon}{1-\gamma}$.

Theorem 4.4 implies the practicality of using hindsight data under condition $g'$ to estimate the expectation. From all illustration above, we give out the final form of the optimization problem for HTRPO:

$$\max_\theta \mathbb{E}_{g'} \left[ \mathbb{E}_{\tau\sim p_\theta(\tau|g)} \left[ \sum_{t=0}^{\infty} \prod_{k=1}^{t} \frac{\pi_{\tilde{\theta}}(a_k|s_k, g')}{\pi_{\tilde{\theta}}(a_k|s_k, g)} \gamma^t \frac{\pi_\theta(a_t|s_t, g')}{\pi_{\tilde{\theta}}(a_t|s_t, g')} A_{\tilde{\theta}}(s_t, a_t, g') \right] \right] \quad (14)$$

$$s.t. \; \mathbb{E}_{g'} \left[ \mathbb{E}_{\tau\sim p_\theta(\tau|g)} \left[ \frac{1}{2} \sum_{t=0}^{\infty} \prod_{k=1}^{t} \frac{\pi_{\tilde{\theta}}(a_k|s_k, g')}{\pi_{\tilde{\theta}}(a_k|s_k, g)} \gamma^t (\log \pi_{\tilde{\theta}}(a_t|s_t, g') - \log \pi_\theta(a_t|s_t, g'))^2 \right] \right] \leq \epsilon'. \quad (15)$$

The solving process for HTRPO optimization problem is explicitly demonstrated in Appendix C and the complete algorithm procedure is included in Appendix D.

## 5 EXPERIMENTS

This section demonstrates the validation of HTRPO on several sparse reward benchmark tasks[1]. The design of our experiments aims to conduct an in-depth investigation in the following aspects:

---

[1]The source code and video can be found at `https://github.com/HTRPOCODES/HTRPO`.

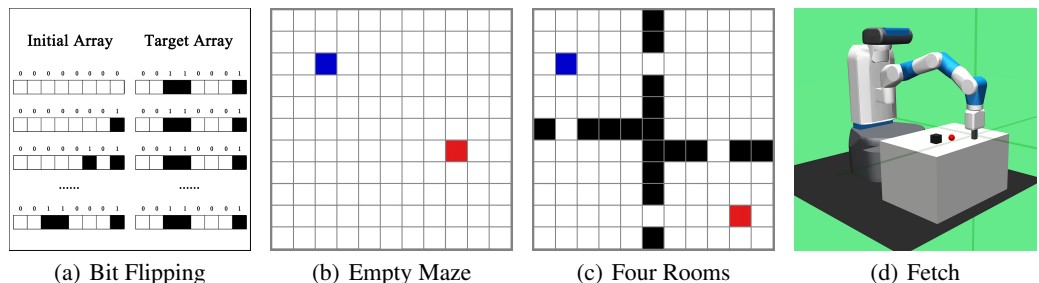

(a) Bit Flipping      (b) Empty Maze      (c) Four Rooms      (d) Fetch

Figure 1: Demonstration of experimental environments

- How is the effectiveness of HTRPO?

- How does each component of HTRPO contribute to its effectiveness?

- How is the performance of policy gradient methods trained with hindsight data in continuous environments?

- How sensitive is HTRPO to network architecture and some key parameters?

## 5.1 EXPERIMENTAL SETTINGS

We implement HTRPO on a variety of reinforcement learning environments, including Bit Flipping, Grid World and Fetch. Among them, Bit Flipping, Grid World, Fetch Reach and Fetch Push are implemented as descrete-action environments while we also conduct continuous version of experiments in Fetch Reach, Fetch Push and Fetch Slide. A glimpse of these environments is demonstrated in Figure 1 while the detailed introductions are included in Appendix F.1. The reward mechanisms are intentionally modified to sparse reward regulations. Besides, for continuous version of Fetch experiments, we apply an additional policy entropy bonus to encourage more exploration.

For each trail of interaction, reward for the agent is set as the remaining number of time steps plus one, and all goals during exploration are chosen uniformly at random for both training and evaluation. During the training process, we terminate one episode either when the maximum number of time steps has elapsed or when the goal state is reached. We evaluate agents' performance by documenting 10 learning trails in the form of average return and their corresponding standard deviation.

In Bit Flipping and Grid World environments, the network architecture is of two hidden layers, each with 64 hyperbolic tangent units; in Fetch environment, for both discrete and continuous implementations, the network contains two 256-unit hidden layers. For all environments mentioned above, we compare HTRPO with HPG (Rauber et al., 2019) and TRPO (Schulman et al., 2015a), which are chosen as the baseline algorithms. Since HPG is never applied to continuous environments in Rauber et al. (2019), we implement HPG to be adapted to continuous environments. Note that the way we scale the time axis is significantly different from that in Rauber et al. (2019). Instead of regarding a certain number of training batches as interval between evaluation steps, we directly uses the accumulated time steps the agent takes while interacting with the environments throughout episodes and batches.

Besides comparing with baselines, we also ablate each component of HTRPO to investigate how significant it is for the final performance. To be specific, we adopt the "vanilla" estimation of KL-divergence which we call "HTRPO with KL1" instead of the proposed one in Section 4; we also observe the performance of our algorithm without weighted importance sampling, which is denoted as "HTRPO without WIS" in this paper.

## 5.2 COMPARATIVE ANALYSIS

In discrete environments, we test both the official version of HPG released in Rauber et al. (2019) and our HPG implementation while for continuous environments of Fetch, we only test our HPG due to the lack of surpport for continuous tasks in Rauber et al. (2019). We apply input normalization in

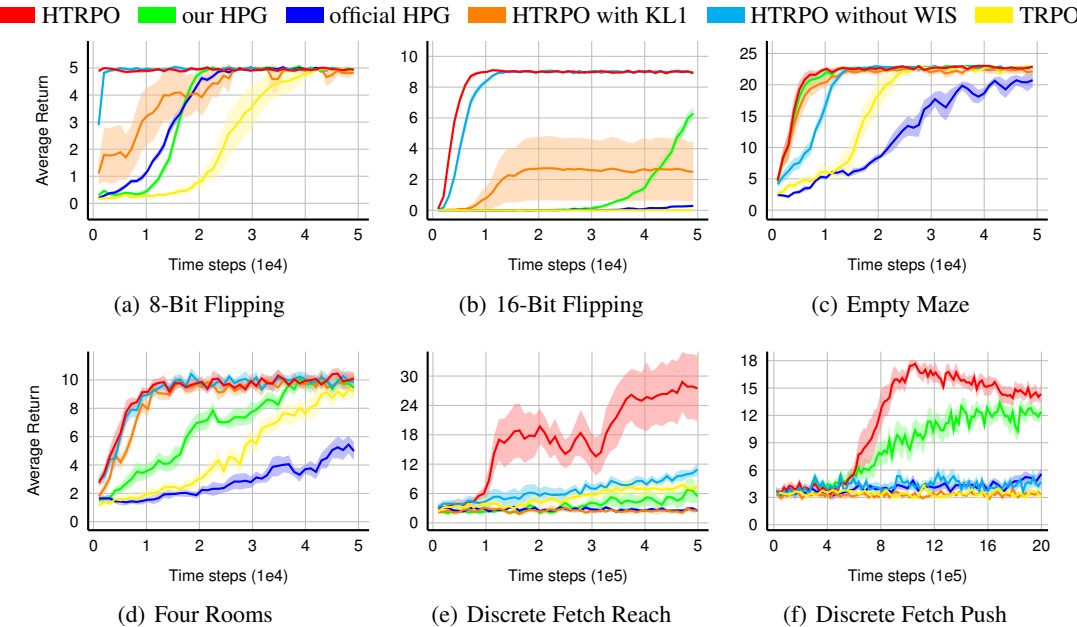

Figure 2: Evaluation curves for discrete environments. The full lines represent the average evaluation over 10 trails and the shaded regions represent the corresponding standard deviation.

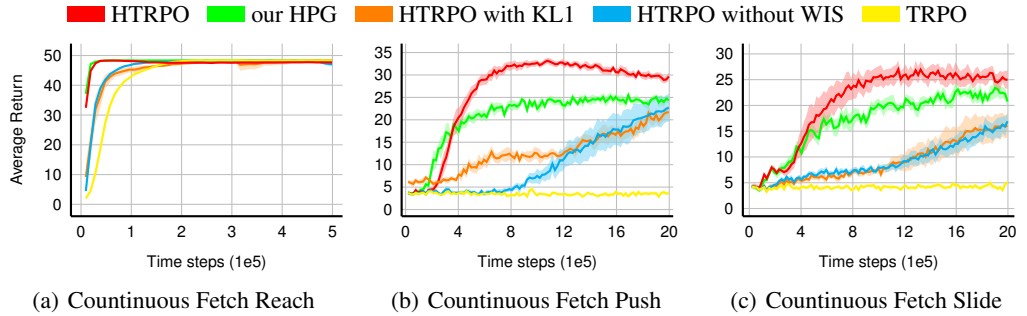

Figure 3: Evaluation curves for continuous environments. The full lines represent the average evaluation over 10 trails and the shaded regions represent the corresponding standard deviation.

the continuous Fetch environments for better performance. However, for fair comparison with the official HPG, we do not employ this trick in any of the discrete environments.

The evaluation curves for the trained policy are demonstrated in Figure 2 and 3 and the training curves and success rate of these experiments are supplemented in Appendix F.3. Detailed settings of hyperparameters for all experiments are listed in Appendix E. From results demonstrated in Rauber et al. (2019), the officially released version of HPG eventually converges to similar performances with that of HTRPO in discrete environments, but sometimes, unlike our HPG, it is still far from converging under this time-step evaluation setting. This kind of distinction in converging speed between our HPG and the official HPG may be caused by the reduction of noises, since we use TD-error to update policies instead of the return corrected by importance sampling, which is adopted in HPG. Thus, for the fairness of comparison, in the following analysis, we mainly compare the properties between HTRPO and our HPG.

**How is the effectiveness of HTRPO?**

From the results we can see that in both discrete and continuous environments, HTRPO outperforms HPG significantly. Aside from assuring a good converging property, the sample efficiency of HTRPO also exceed that of HPG, for it reaches a higher average return within less time in most

environments. As for TRPO, though it can converge in several simple tasks like Bit Flipping, Grid World and continuous Fetch Reach, it remains incompetent in dealing with complex control tasks including Fetch Push and Fetch Slide, in all of which HTRPO can learn a good policy. The reason is that for TRPO, it is basically impossible to acquire a positive reward at the beginning of the training in such environments, which makes the policy updates meaningless.

**How does each component of HTRPO contribute to its effectiveness?**

In both Figure 2 and Figure 3, "HTRPO with KL1" and "HTRPO without WIS" performs much worse than the complete version of HTRPO. When we estimate the KL-divergence using the "vanilla" KL-divergence defined as equation 9, it causes severe instability, which means that the estimated KL-divergence can be negative with an unacceptable probability. Considering the practicality of the experiment, the corresponding iteration will be skipped without any updates of the policy in this senario. Given the phenomenon stated above, the final performance of "HTRPO with KL1" is much worse and more unstable in all environments. As for the study of Weighted Importance Sampling, it is widely known for significantly reducing the variance (Bishop, 2016), which is once again proved by the results of "HTRPO without WIS". Admittedly, we can see that the performance of "HTRPO without WIS" matches the full version of HTRPO in several simple environments in Figure 2 (a)-(d) and Figure 3 (a). However, for more complex environments like Fetch Push and Fetch Slide, the variance is detrimentally larger than that in simple environments. In short, the performance of "HTRPO without WIS" has a severe degradation comparing to the full version of HTRPO.

**How is the performance of policy gradient methods trained with hindsight data in continuous environments?**

As mentioned in Plappert et al. (2018), it still remains unexplored that to what extent the policy gradient methods trained with hindsight data can solve continuous control tasks. In this section, we will provide the answer. We implement HTRPO in continuous control tasks including Fetch Reach, Fetch Push and Fetch Slide. HPG is tested as well for comparison. From the results, we can see that with the help of input normalization, HPG can learn a valid policy in continuous control tasks. Still, HTRPO performs much better than HPG in all three environments, benefiting from a faster and more stable convergence. As illustrated in Figure 3, HTRPO eventually achieves an average success rate of 92% for Fetch Push and 82.5% for Fetch Slide.

**How sensitive is HTRPO to network architecture and some key parameters?**

To study the sensitivity of HTRPO to different network architectures, we observe the performance of HTRPO with different network settings. From the results demonstrated in Appendix F.2.1, HTRPO achieves commendable performances with all three different network architectures while HPG only converges under certain settings. As for the sensitivity of HTRPO to key parameters, we mainly observe the impact of different number of alternative goals. Based on the learning curves in Appendix F.2.2, we can see that Hindishgt TRPO with more alternative goals achieves better converging speed.

## 6 CONCLUSION

We have extended the monotonically converging on-policy algorithm TRPO to accommodate sparse reward environments by adopting the hindsight methodology. The optimization problem in TRPO is scrupulously derived into hindsight formulation and, when the KL-divergence in the constraint function is small enough, it can be tactfully approximated by another $f$-divergence in order to reduce estimation variance and improve learning stability. Experimental results on a variety of environments demonstrate effective performances of HTRPO, and validate its sample efficiency and stable policy update quality in both discrete and continuous scenarios. Therefore, this work reveals HTRPO's vast potential in solving sparse reward reinforcement learning problems.

ACKNOWLEDGMENTS

We greatly acknowledge all the fundings in support of this work.

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

# A  PROOF FOR OBJECTIVE FUNCTION

## A.1  DERIVIATION FROM EQUATION 3 TO EQUATION 5

With no influence to the optimal solution, we can multiply equation 3 by a constant $\frac{1}{1-\gamma}$,

$$
\begin{aligned}
L_{\tilde{\theta}}(\theta) &= \frac{1}{1-\gamma} \underset{s\sim\rho_{\tilde{\theta}}, a\sim\pi_{\tilde{\theta}}(a|s)}{\mathbb{E}} \left[ \frac{\pi_\theta(a|s)}{\pi_{\tilde{\theta}}(a|s)} A_{\tilde{\theta}}(s,a) \right] \\
&= \frac{1}{1-\gamma} \sum_{s\in\mathcal{S}} \frac{\sum_{t=0}^{\infty}\gamma^t P(s_t=s)}{\frac{1}{1-\gamma}} \underset{a\sim\pi_{\tilde{\theta}}(a|s)}{\mathbb{E}} \left[ \frac{\pi_\theta(a|s)}{\pi_{\tilde{\theta}}(a|s)} A_{\tilde{\theta}}(s,a) \right] \\
&= \sum_{t=0}^{\infty} \gamma^t \underset{s_t\sim p_{\tilde{\theta}}(s_t), a_t\sim\pi_{\tilde{\theta}}(a_t|s_t)}{\mathbb{E}} \left[ \frac{\pi_\theta(a_t|s_t)}{\pi_{\tilde{\theta}}(a_t|s_t)} A_{\tilde{\theta}}(s_t,a_t) \right] \\
&= \underset{\tau\sim p_{\tilde{\theta}}(\tau)}{\mathbb{E}} \left[ \sum_{t=0}^{\infty} \gamma^t \frac{\pi_\theta(a_t|s_t)}{\pi_{\tilde{\theta}}(a_t|s_t)} A_{\tilde{\theta}}(s_t,a_t) \right]
\end{aligned}
\tag{16}
$$

## A.2  THEOREM 3.1

**Theorem 3.1** (HTRPO Objective Function). For the original goal $g$ and an alternative goal $g'$, the object function of HTRPO $L_{\tilde{\theta}}(\theta)$ is given by:

$$
L_{\tilde{\theta}}(\theta) = \underset{g'}{\mathbb{E}} \left[ \underset{\tau\sim p_\theta(\tau|g)}{\mathbb{E}} \left[ \sum_{t=0}^{\infty} \prod_{k=1}^{t} \frac{\pi_{\tilde{\theta}}(a_k|s_k,g')}{\pi_{\tilde{\theta}}(a_k|s_k,g)} \gamma^t \frac{\pi_\theta(a_t|s_t,g')}{\pi_{\tilde{\theta}}(a_t|s_t,g')} A_{\tilde{\theta}}(s_t,a_t,g') \right] \right],
\tag{7}
$$

in which, $\tau = s_1, a_1, s_2, a_2, ..., s_t, a_t$.

*Proof.* Starting from equation 6, for every time step $t$ in the expectation, denote

$$
L_{\tilde{\theta}}(\theta, t) = \underset{g}{\mathbb{E}} \left[ \underset{\tau\sim p_{\tilde{\theta}}(\tau|g)}{\mathbb{E}} \left[ \gamma^t \frac{\pi_\theta(a_t|s_t,g)}{\pi_{\tilde{\theta}}(a_t|s_t,g)} A_{\tilde{\theta}}(s_t,a_t,g) \right] \right],
\tag{17}
$$

so that

$$
L_{\tilde{\theta}}(\theta) = \sum_{t=0}^{\infty} L_{\tilde{\theta}}(\theta, t).
\tag{18}
$$

Split every trajectory $\tau$ into $\tau_1$ and $\tau_2$ where $\tau_1 = s_1, a_1, s_2, a_2, ..., s_t, a_t$ and $\tau_2 = s_{t+1}, a_{t+1}, ...,$ then

$$
L_{\tilde{\theta}}(\theta, t) = \underset{g}{\mathbb{E}} \left[ \underset{\tau_1\sim p_{\tilde{\theta}}(\tau_1|g)}{\mathbb{E}} \left[ \underset{\tau_2\sim p_{\tilde{\theta}}(\tau_2|\tau_1,g)}{\mathbb{E}} \left[ \gamma^t \frac{\pi_\theta(a_t|s_t,g)}{\pi_{\tilde{\theta}}(a_t|s_t,g)} A_{\tilde{\theta}}(s_t,a_t,g) \right] \right] \right].
\tag{19}
$$

For that $\gamma^t \frac{\pi_\theta(a_t|s_t,g)}{\pi_{\tilde{\theta}}(a_t|s_t,g)} A_{\tilde{\theta}}(s_t,a_t,g)$ is independent from $\tau_2$ conditioned on $\tau_1$,

$$
\begin{aligned}
L_{\tilde{\theta}}(\theta, t) &= \underset{g}{\mathbb{E}} \left[ \underset{\tau_1\sim p_{\tilde{\theta}}(\tau_1|g)}{\mathbb{E}} \left[ \gamma^t \frac{\pi_\theta(a_t|s_t,g)}{\pi_{\tilde{\theta}}(a_t|s_t,g)} A_{\tilde{\theta}}(s_t,a_t,g) \right] \underset{\tau_2\sim p_{\tilde{\theta}}(\tau_2|\tau_1,g)}{\mathbb{E}} [1] \right] \\
&= \underset{g}{\mathbb{E}} \left[ \underset{\tau_1\sim p_{\tilde{\theta}}(\tau_1|g)}{\mathbb{E}} \left[ \gamma^t \frac{\pi_\theta(a_t|s_t,g)}{\pi_{\tilde{\theta}}(a_t|s_t,g)} A_{\tilde{\theta}}(s_t,a_t,g) \right] \right] \\
&= \underset{g}{\mathbb{E}} \left[ \underset{s_{1:t},a_{1:t}\sim p_{\tilde{\theta}}(s_{1:t},a_{1:t}|g)}{\mathbb{E}} \left[ \gamma^t \frac{\pi_\theta(a_t|s_t,g)}{\pi_{\tilde{\theta}}(a_t|s_t,g)} A_{\tilde{\theta}}(s_t,a_t,g) \right] \right]
\end{aligned}
\tag{20}
$$

Thus,

$$
L_{\tilde{\theta}}(\theta) = \sum_{t=0}^{\infty} \underset{g}{\mathbb{E}} \left[ \underset{s_{1:t},a_{1:t}\sim p_{\tilde{\theta}}(s_{1:t},a_{1:t}|g)}{\mathbb{E}} \left[ \gamma^t \frac{\pi_\theta(a_t|s_t,g)}{\pi_{\tilde{\theta}}(a_t|s_t,g)} A_{\tilde{\theta}}(s_t,a_t,g) \right] \right]
\tag{21}
$$

Following the techniques of importance sampling, the objective function can be rewritten in the form of new goal $g'$ :

$$L_{\tilde{\theta}}(\theta) = \sum_{t=0}^{\infty} \mathbb{E}_{g'} \left[ \mathbb{E}_{s_{1:t},a_{1:t} \sim p_{\tilde{\theta}}(s_{1:t},a_{1:t}|g)} \left[ \frac{p_{\tilde{\theta}}(s_{1:t},a_{1:t}|g')}{p_{\tilde{\theta}}(s_{1:t},a_{1:t}|g)} \gamma^t \frac{\pi_\theta(a_t|s_t,g)}{\pi_{\tilde{\theta}}(a_t|s_t,g)} A_{\tilde{\theta}}(s_t,a_t,g') \right] \right]. \quad (22)$$

Furthermore, given that

$$p(s_{1:t},a_{1:t}|g) = p(s_1)p(a_t|s_t,g) \prod_{k=1}^{t-1} p(a_k|s_k,g)p(s_{k+1}|s_k,a_k), \quad (23)$$

after expanding the objective function and cancelling terms,

$$L_{\tilde{\theta}}(\theta) = \sum_{t=0}^{\infty} \mathbb{E}_{g'} \left[ \mathbb{E}_{s_{1:t},a_{1:t} \sim p_{\tilde{\theta}}(s_{1:t},a_{1:t}|g)} \left[ \prod_{k=1}^{t} \frac{\pi_{\tilde{\theta}}(a_k|s_k,g')}{\pi_{\tilde{\theta}}(a_k|s_k,g)} \gamma^t \frac{\pi_\theta(a_t|s_t,g')}{\pi_{\tilde{\theta}}(a_t|s_t,g')} A_{\tilde{\theta}}(s_t,a_t,g') \right] \right]$$

$$(24)$$

### A.3   Theorem 3.2

**Theorem 3.2** (Gradient of HTRPO Objective Function). For the original goal $g$ and an alternative goal $g'$, the gradient $\nabla_\theta L_{\tilde{\theta}}(\theta)$ of HTRPO object function with respect to $\theta$ is given by the following expression:

$$\nabla_\theta L_{\tilde{\theta}}(\theta) = \mathbb{E}_{g'} \left[ \mathbb{E}_{\tau \sim p_\theta(\tau|g))} \left[ \sum_{t=0}^{\infty} \prod_{k=1}^{t} \frac{\pi_{\tilde{\theta}}(a_k|s_k,g')}{\pi_{\tilde{\theta}}(a_k|s_k,g)} \gamma^t \frac{\nabla_\theta \pi_\theta(a_t|s_t,g')}{\pi_{\tilde{\theta}}(a_t|s_t,g')} A_{\tilde{\theta}}(s_t,a_t,g') \right] \right], \quad (8)$$

in which $\tau = s_1,a_1,s_2,a_2,...,s_t,a_t$.

*Proof.* Starting from equation 24, since that $\pi_\theta(a_t|s_t,g')$ is the only term relavant to $\theta$, the corresponding gradient of the objective function is computed by:

$$\nabla_\theta L_{\tilde{\theta}}(\theta) = \sum_{t=0}^{\infty} \mathbb{E}_{g'} \left[ \mathbb{E}_{s_{1:t},a_{1:t} \sim p_{\tilde{\theta}}(s_{1:t},a_{1:t}|g)} \left[ \prod_{k=1}^{t} \frac{\pi_{\tilde{\theta}}(a_k|s_k,g')}{\pi_{\tilde{\theta}}(a_k|s_k,g)} \gamma^t \frac{\nabla_\theta \pi_\theta(a_t|s_t,g')}{\pi_{\tilde{\theta}}(a_t|s_t,g')} A_{\tilde{\theta}}(s_t,a_t,g') \right] \right].$$

$$(25)$$

# B  PROOF FOR CONSTRAINT FUNCTION

## B.1  THEOREM 4.1

**Theorem 4.1** (Logarithmic Form of Constraint Function). Given two policies $\pi_{\tilde{\theta}}(a|s)$ and $\pi_\theta(a|s)$, the expectation of their KL-divergence over states $s \sim \rho_{\tilde{\theta}}(s)$ is written as:

$$\mathop{\mathbb{E}}_{s\sim\rho_{\tilde{\theta}}(s)} \left[ D_{KL}(\pi_{\tilde{\theta}}(a|s)||\pi_\theta(a|s)) \right] = \mathop{\mathbb{E}}_{s,a\sim\rho_{\tilde{\theta}}(s,a)} \left[ \log \pi_{\tilde{\theta}}(a|s) - \log \pi_\theta(a|s) \right] \qquad (9)$$

*Proof.* Expand the expectation in equation 2 by the definition of KL-divergence,

$$\mathop{\mathbb{E}}_{s\sim\rho_{\tilde{\theta}}(s)} \left[ D_{KL}(\pi_{\tilde{\theta}}(a|s)||\pi_\theta(a|s)) \right] = \mathop{\mathbb{E}}_{s\sim\rho_{\tilde{\theta}}(s)} \left[ \mathop{\mathbb{E}}_{a\sim\pi_{\tilde{\theta}}(a|s)} \left[ \log \frac{\pi_{\tilde{\theta}}(a|s)}{\pi_\theta(a|s)} \right] \right]$$

$$= \mathop{\mathbb{E}}_{s\sim\rho_{\tilde{\theta}}(s),a\sim\pi_{\tilde{\theta}}(a|s)} \left[ \log \pi_{\tilde{\theta}}(a|s) - \log \pi_\theta(a|s) \right] \qquad (26)$$

## B.2  THEOREM 4.2

**Lemma B.1** Given two distibutions $p(x)$ and $q(x)$, $q(x) = p(x) + \eta(x)$, in which $\eta(x)$ is the variation of $q(x)$ at $p(x)$.

$$\mathbb{E}\left[\log p(x) - \log q(x)\right] = \mathbb{E}\left[\frac{1}{2}(\log p(x) - \log q(x))^2\right] + \mathbb{E}\left[o(\eta(x)^3)\right] \qquad (27)$$

*Proof.* Consider the second order Taylor expansion of $\log q(x)$ at $p(x)$,

$$\log q(x) = \log p(x) + \frac{1}{p(x)}\eta(x) - \frac{1}{2p(x)^2}\eta(x)^2 + o(\eta(x)^3) \qquad (28)$$

For the left side of equation 27,

$$\mathbb{E}\left[\log p(x) - \log q(x)\right] = \mathbb{E}\left[-\frac{1}{p(x)}\eta(x) + \frac{1}{2p(x)^2}\eta(x)^2 - o(\eta(x)^3)\right]$$

$$= \int (-\frac{1}{p(x)}\eta(x) + \frac{1}{2p(x)^2}\eta(x)^2 - o(\eta(x)^3))p(x)\,dx$$

$$= \int (\frac{1}{2p(x)}\eta(x)^2 - p(x)o(\eta(x)^3))\,dx. \qquad (29)$$

For the first term on the right side of equation 27,

$$\mathbb{E}\left[\frac{1}{2}(\log p(x) - \log q(x))^2\right] = \frac{1}{2}\mathbb{E}\left[(-\frac{1}{p(x)}\eta(x) + \frac{1}{2p(x)^2}\eta(x)^2 - o(\eta(x)^3))^2\right]$$

$$= \frac{1}{2}\mathbb{E}\left[\frac{1}{p(x)^2}\eta(x)^2 + o(\eta(x)^3)\right]$$

$$= \frac{1}{2}\int (\frac{1}{p(x)^2}\eta(x)^2 + o(\eta(x)^3))p(x)\,dx$$

$$= \int (\frac{1}{2p(x)}\eta(x)^2 + \frac{1}{2}p(x)o(\eta(x)^3))\,dx$$

$$= \int (\frac{1}{2p(x)}\eta(x)^2 - p(x)o(\eta(x)^3))\,dx + \int p(x)o(\eta(x)^3)\,dx$$

$$= \mathbb{E}\left[\log p(x) - \log q(x)\right] + \mathbb{E}\left[o(\eta(x)^3)\right]. \qquad (30)$$

**Theorem 4.2** (Approximation of Constraint Function). For policy $\pi_{\tilde{\theta}}(a|s)$ and $\pi_\theta(a|s)$, and for $\eta = \pi_\theta(a|s) - \pi_{\tilde{\theta}}(a|s)$,

$$\mathop{\mathbb{E}}_{s,a\sim\rho_{\tilde{\theta}}(s,a)} \left[ \log \pi_{\tilde{\theta}}(a|s) - \log \pi_\theta(a|s) \right] = \mathop{\mathbb{E}}_{s,a\sim\rho_{\tilde{\theta}}(s,a)} \left[ \frac{1}{2}(\log \pi_{\tilde{\theta}}(a|s) - \log \pi_\theta(a|s))^2 \right]$$

$$+ \mathop{\mathbb{E}}_{s,a\sim\rho_{\tilde{\theta}}(s,a)} \left[ o(\eta^3) \right]. \qquad (10)$$

*Proof.* Based on Lemma A.1, let $p(x) = \pi_{\tilde{\theta}}(a|s)$ and $q(x) = \pi_\theta(a|s)$, equation 27 results in equation 10.

### B.3 THEOREM 4.3

**Lemma B.2** For any random variable $Y \in [0, 0.5]$,

$$\text{Var}(Y^2) \le \text{Var}(Y), \tag{31}$$

in which $\text{Var}(Y)$ denotes the variance of $Y$.

*Proof.*

$$
\begin{aligned}
\text{Var}(Y) - \text{Var}(Y^2) &= \mathbb{E}(Y^2) - [\mathbb{E}(Y)]^2 - \mathbb{E}(Y^4) + \left[\mathbb{E}(Y^2)\right]^2 \\
&= \left[\mathbb{E}(Y^2) - \mathbb{E}(Y^4)\right] - \left[[\mathbb{E}(Y)]^2 - \left[\mathbb{E}(Y^2)\right]^2\right] \\
&= \mathbb{E}\left[Y^2(1-Y)(1+Y)\right] - \mathbb{E}\left[Y(1-Y)\right]\mathbb{E}\left[Y(1+Y)\right] \\
&= \text{Cov}(Y(1+Y), Y(1-Y)). 
\end{aligned}
\tag{32}
$$

Denote $X_1(Y) = Y(1+Y)$ and $X_2(Y) = Y(1-Y)$. Then,

$$
\begin{aligned}
\text{Var}(Y) - \text{Var}(Y^2) &= \text{Cov}(X_1, X_2) \\
&= \mathbb{E}\left[X_1\left(X_2 - \mathbb{E}(X_2)\right)\right]
\end{aligned}
\tag{33}
$$

There always exists $Y_0 \in [0, 0.5]$ that saticefies $X_2(Y_0) = \mathbb{E}(X_2)$. When $Y = Y_0$, let $X_1(Y_0) = \mu_1$ in which $\mu_1$ is a constant. Then the equation can be converted by the following steps:

$$
\begin{aligned}
\text{Var}(Y) - \text{Var}(Y^2) &= \mathbb{E}\left[(X_1 - \mu_1)\left(X_2 - \mathbb{E}(X_2)\right)\right] + \mu_1\mathbb{E}\left[X_2 - \mathbb{E}(X_2)\right] \\
&= \mathbb{E}\left[(X_1 - \mu_1)\left(X_2 - \mathbb{E}(X_2)\right)\right]
\end{aligned}
\tag{34}
$$

Thus, when $Y = Y_0$, the two factors in equation 34, $(X_1 - \mu_1)$ and $(X_2 - \mathbb{E}(X_2))$ equal to 0 simultaneously.

Also, it is easy to notice that when $Y \in [0, 0.5]$, $X_1$ and $X_2$ are strictly increasing with the increase of $Y$. Thus, $(X_1 - \mu_1)$ and $(X_2 - \mathbb{E}(X_2))$ are either both positive or both negative, if not zero. Therefore,

$$\text{Var}(Y) - \text{Var}(Y^2) \ge 0. \tag{35}$$

**Lemma B.3** For any random variable Y,

$$\text{Var}(|Y|) \le \text{Var}(Y), \tag{36}$$

in which $\text{Var}(Y)$ denotes the variance of $Y$.

*Proof.* Apparently,

$$\int_y |f(y)|\, dy \ge |\int_y f(y)\, dy|. \tag{37}$$

Consequently,

$$\mathbb{E}(|Y|) \ge |\mathbb{E}(Y)|. \tag{38}$$

For that

$$\text{Var}(Y) = \mathbb{E}(Y^2) - (\mathbb{E}(Y))^2, \tag{39}$$

we have

$$\text{Var}(|Y|) \le \text{Var}(Y). \tag{36}$$

**Theorem 4.3** (Variance of Constraint Function). For policy $\pi_{\tilde{\theta}}(a|s)$ and $\pi_{\theta}(a|s)$, let Var denotes the variance of a variable. When $\log \pi_{\tilde{\theta}}(a|s) - \log \pi_{\theta}(a|s) \in [-0.5, 0.5]$, then

$$\operatorname*{Var}_{s,a\sim\rho_{\tilde{\theta}}(s,a)} \left[ \frac{(\log \pi_{\tilde{\theta}}(a|s) - \log \pi_{\theta}(a|s))^2}{2} \right] < \operatorname*{Var}_{s,a\sim\rho_{\tilde{\theta}}(s,a)} \left[ \log \pi_{\tilde{\theta}}(a|s) - \log \pi_{\theta}(a|s) \right]. \tag{11}$$

*Proof.* let $Y = |\log \pi_{\tilde{\theta}}(a|s) - \log \pi_{\theta}(a|s)|$. Given Lemma B.2,

$$\operatorname*{Var}_{s,a\sim\rho_{\tilde{\theta}}(s,a)} \left[ |\log \pi_{\tilde{\theta}}(a|s) - \log \pi_{\theta}(a|s)|^2 \right] \leq \operatorname*{Var}_{s,a\sim\rho_{\tilde{\theta}}(s,a)} \left[ |\log \pi_{\tilde{\theta}}(a|s) - \log \pi_{\theta}(a|s)| \right] \tag{40}$$

Given Lemma B.3,

$$\operatorname*{Var}_{s,a\sim\rho_{\tilde{\theta}}(s,a)} \left[ |\log \pi_{\tilde{\theta}}(a|s) - \log \pi_{\theta}(a|s)| \right] \leq \operatorname*{Var}_{s,a\sim\rho_{\tilde{\theta}}(s,a)} \left[ \log \pi_{\tilde{\theta}}(a|s) - \log \pi_{\theta}(a|s) \right]. \tag{41}$$

With the transitivity of inequality, combining equation 40 and equation 41, we know that

$$\operatorname*{Var}_{s,a\sim\rho_{\tilde{\theta}}(s,a)} \left[ |\log \pi_{\tilde{\theta}}(a|s) - \log \pi_{\theta}(a|s)|^2 \right] \leq \operatorname*{Var}_{s,a\sim\rho_{\tilde{\theta}}(s,a)} \left[ \log \pi_{\tilde{\theta}}(a|s) - \log \pi_{\theta}(a|s) \right]. \tag{42}$$

### B.4 THEOREM 4.4

**Theorem 4.4** (HTRPO Constraint Function). For the original goal $g$ and an alternative goal $g'$, the constraint between policy $\pi_{\tilde{\theta}}(a|s)$ and policy $\pi_{\theta}(a|s)$ is given by:

$$\mathbb{E}_{g'} \left[ \mathbb{E}_{\tau\sim p_{\theta}(\tau|g)} \left[ \frac{1}{2} \sum_{t=0}^{\infty} \prod_{k=1}^{t} \frac{\pi_{\tilde{\theta}}(a_k|s_k, g')}{\pi_{\tilde{\theta}}(a_k|s_k, g)} \gamma^t (\log \pi_{\tilde{\theta}}(a_t|s_t, g') - \log \pi_{\theta}(a_t|s_t, g'))^2 \right] \right] \leq \epsilon', \tag{13}$$

in which $\epsilon' = \frac{\epsilon}{1-\gamma}$.

*Proof.* Starting from equation 9, the constraint condition is written as:

$$\mathbb{E}_{s,a\sim\rho_{\tilde{\theta}}(s,a)} \left[ \log \pi_{\tilde{\theta}}(a|s) - \log \pi_{\theta}(a|s) \right] \leq \epsilon \tag{43}$$

Given Theorem 4.2,

$$\mathbb{E}_{s,a\sim\rho_{\tilde{\theta}}(s,a)} \left[ \frac{1}{2} (\log \pi_{\tilde{\theta}}(a|s) - \log \pi_{\theta}(a|s))^2 \right] \leq \epsilon \tag{44}$$

Multiply the constraint function by a constant $\frac{1}{1-\gamma}$,

$$\frac{1}{1-\gamma} \mathbb{E}_{s,a\sim\rho_{\tilde{\theta}}(s,a)} \left[ \frac{1}{2} (\log \pi_{\tilde{\theta}}(a|s) - \log \pi_{\theta}(a|s))^2 \right] \leq \frac{\epsilon}{1-\gamma} \tag{45}$$

Denote the constraint function as $f_{\tilde{\theta}}(\theta)$,

$$\begin{aligned}
f_{\tilde{\theta}}(\theta) &= \frac{1}{1-\gamma} \sum_{s\in\mathcal{S}} \frac{\sum_{t=0}^{\infty} \gamma^t P(s_t = s)}{\frac{1}{1-\gamma}} \mathbb{E}_{a\sim\pi_{\tilde{\theta}}(a|s)} \left[ \frac{1}{2} (\log \pi_{\tilde{\theta}}(a|s) - \log \pi_{\theta}(a|s))^2 \right] \\
&= \sum_{t=0}^{\infty} \gamma^t \mathbb{E}_{s_t\sim p_{\tilde{\theta}}(s_t), a_t\sim\pi_{\tilde{\theta}}(a_t|s_t)} \left[ \frac{1}{2} (\log \pi_{\tilde{\theta}}(a_t|s_t) - \log \pi_{\theta}(a_t|s_t))^2 \right] \\
&= \mathbb{E}_{\tau\sim p_{\tilde{\theta}}(\tau)} \left[ \frac{1}{2} \sum_{t=0}^{\infty} \gamma^t (\log \pi_{\tilde{\theta}}(a_t|s_t) - \log \pi_{\theta}(a_t|s_t))^2 \right]
\end{aligned} \tag{46}$$

To write the constraint function in goal-conditioned form,

$$f_{\tilde{\theta}}(\theta) = \mathbb{E}_{g} \mathbb{E}_{\tau\sim p_{\tilde{\theta}}(\tau|g)} \left[ \frac{1}{2} \sum_{t=0}^{\infty} \gamma^t (\log \pi_{\tilde{\theta}}(a_t|s_t, g) - \log \pi_{\theta}(a_t|s_t, g))^2 \right] \tag{47}$$

In a similar way with the proof for Theorem 3.1, denote every time step of $f_{\tilde{\theta}}(\theta)$ as $f_{\tilde{\theta}}(\theta, t)$, in other words,

$$f_{\tilde{\theta}}(\theta) = \sum_{t=0}^{\infty} f_{\tilde{\theta}}(\theta, t) \tag{48}$$

for trajectory $\tau_1 = s_1, a_1, s_2, a_2, ..., s_t, a_t$ and $\tau_2 = s_{t+1}, a_{t+1}, ...$,

$$f_{\tilde{\theta}}(\theta, t) = \mathbb{E}_{g} \left[ \mathbb{E}_{\tau_1 \sim p_{\tilde{\theta}}(\tau_1|g)} \left[ \mathbb{E}_{\tau_2 \sim p_{\tilde{\theta}}(\tau_2|\tau_1, g)} \left[ \frac{1}{2} \gamma^t (\log \pi_{\tilde{\theta}}(a_t|s_t, g) - \log \pi_{\theta}(a_t|s_t, g))^2 \right] \right] \right]. \tag{49}$$

For that $\frac{1}{2}(\log \pi_{\tilde{\theta}}(a_t|s_t, g) - \log \pi_{\theta}(a_t|s_t, g))^2$ is independent from $\tau_2$ conditioned on $\tau_1$,

$$f_{\tilde{\theta}}(\theta, t) = \mathbb{E}_{g} \left[ \mathbb{E}_{\tau_1 \sim p_{\tilde{\theta}}(\tau_1|g)} \left[ \frac{1}{2} \gamma^t (\log \pi_{\tilde{\theta}}(a_t|s_t, g) - \log \pi_{\theta}(a_t|s_t, g))^2 \right] \mathbb{E}_{\tau_2 \sim p_{\tilde{\theta}}(\tau_2|g)} [1] \right]$$

$$= \mathbb{E}_{g} \left[ \mathbb{E}_{\tau_1 \sim p_{\tilde{\theta}}(\tau_1|g)} \left[ \frac{1}{2} \gamma^t (\log \pi_{\tilde{\theta}}(a_t|s_t, g) - \log \pi_{\theta}(a_t|s_t, g))^2 \right] \right]. \tag{50}$$

Accordingly,

$$f_{\tilde{\theta}}(\theta) = \sum_{t=0}^{\infty} \mathbb{E}_{g} \left[ \mathbb{E}_{s_{1:t}, a_{1:t} \sim p_{\tilde{\theta}}(s_{1:t}, a_{1:t}|g)} \left[ \frac{1}{2} \gamma^t (\log \pi_{\tilde{\theta}}(a_t|s_t, g) - \log \pi_{\theta}(a_t|s_t, g))^2 \right] \right]. \tag{51}$$

Furthermore, by importance sampling, for a new goal $g'$, the constraint can be converted to the following form

$$f_{\tilde{\theta}}(\theta) = \mathbb{E}_{g'} \left[ \mathbb{E}_{\tau \sim p_{\theta}(\tau|g)} \left[ \frac{1}{2} \sum_{t=0}^{\infty} \frac{p_{\tilde{\theta}}(s_{1:t}, a_{1:t}|g')}{p_{\tilde{\theta}}(s_{1:t}, a_{1:t}|g)} \gamma^t (\log \pi_{\tilde{\theta}}(a_t|s_t, g') - \log \pi_{\theta}(a_t|s_t, g'))^2 \right] \right]. \tag{52}$$

in which $\tau = s_1, a_1, s_2, a_2, ..., s_t, a_t$. Denote $\epsilon' = \frac{\epsilon}{1-\gamma}$. Based on equation 23, by expanding and canceling terms, the constraint condition can be written as

$$\mathbb{E}_{g'} \left[ \mathbb{E}_{\tau \sim p_{\theta}(\tau|g)} \left[ \frac{1}{2} \sum_{t=0}^{\infty} \prod_{k=1}^{t} \frac{\pi_{\tilde{\theta}}(a_k|s_k, g')}{\pi_{\tilde{\theta}}(a_k|s_k, g)} \gamma^t (\log \pi_{\tilde{\theta}}(a_t|s_t, g') - \log \pi_{\theta}(a_t|s_t, g'))^2 \right] \right] \leq \epsilon'. \tag{13}$$

## C  SOLVING PROCESS FOR HTRPO

### C.1  HTRPO ESTIMATORS

Based on the final form of HTRPO optimization problem, this section completes the feasibility of this algorithm with estimators for the objective function and the KL-divergence constraint.

Given a dataset of trajectories and goals $\mathcal{D} = \{\boldsymbol{\tau}^{(i)}, g^{(i)}\}_{i=1}^N$, each trajectory $\boldsymbol{\tau}^{(i)}$ is obtained from interacting with the environment under a goal $g^{(i)}$. In order to generate hindsight experience, we also need to sample a set of alternative goals $\mathcal{G} = \{g'^{(i)}\}_{i=1}^{N_g}$. The Monte Carlo estimation of HTRPO optimization problem with dataset $\mathcal{D}$ can be derived as follows:

$$\max_\theta \frac{1}{\lambda} \sum_{g' \in \mathcal{G}} \sum_{i=1}^N \sum_{t=0}^\infty \left[ \prod_{k=1}^t \frac{\pi_{\tilde{\theta}}(a_k^{(i)}|s_k^{(i)}, g')}{\pi_{\tilde{\theta}}(a_k^{(i)}|s_k^{(i)}, g^{(i)})} \gamma^t \frac{\nabla_\theta \pi_\theta(a_k^{(i)}|s_k^{(i)}, g')}{\pi_{\tilde{\theta}}(a_k^{(i)}|s_k^{(i)}, g')} A_{\tilde{\theta}}(a_k^{(i)}, s_k^{(i)}, g') \right] \tag{53}$$

$$s.t. \frac{1}{\lambda} \sum_{g' \in \mathcal{G}} \sum_{i=1}^N \sum_{t=0}^\infty \left[ \frac{\gamma^t}{2} \prod_{k=1}^t \frac{\pi_{\tilde{\theta}}(a_k^{(i)}|s_k^{(i)}, g')}{\pi_{\tilde{\theta}}(a_k^{(i)}|s_k^{(i)}, g^{(i)})} (\log \pi_{\tilde{\theta}}(a_k^{(i)}|s_k^{(i)}, g') - \log \pi_\theta(a_k^{(i)}|s_k^{(i)}, g'))^2 \right] \leq \epsilon', \tag{54}$$

in which $\lambda = N \cdot N_g$ and $g'$ is supposed to follow uniform distribution. However, in experiments, we follow the alternative goal sampling method of HPG (Rauber et al., 2019). As a result, the goals of training data actually follow the distribution of alternative goals instead of uniform distribution, and the objective and KL expectation will be estimated w.r.t. the alternative goal distribution. Therefore, during the learning process, our algorithm is encouraging the agent to achieve the alternative goals. Such mechanism serves as a mutual approach for all hindsight methods (Andrychowicz et al., 2017; Rauber et al., 2019; Plappert et al., 2018), which can be seen as a merit, for the intention is to guide the agent to achieve the alternative goals and then generalize to the original goals.

However, as discussed in Rauber et al. (2019), this kind of estimation may result in excessive variance, which leads to an unstable learning curve. In order to avoid instability, we adopt the technique of weighted importance sampling introduced in Bishop (2016) and further convert the optimization problem to the following form:

$$\max_\theta \frac{1}{\lambda} \sum_{g' \in \mathcal{G}} \sum_{i=1}^N \sum_{t=0}^\infty \left[ \frac{\prod_{k=1}^t \frac{\pi_{\tilde{\theta}}(a_k^{(i)}|s_k^{(i)}, g')}{\pi_{\tilde{\theta}}(a_k^{(i)}|s_k^{(i)}, g^{(i)})}}{\sum_{j=1}^N \prod_{k=1}^t \frac{\pi_{\tilde{\theta}}(a_k^{(i)}|s_k^{(i)}, g')}{\pi_{\tilde{\theta}}(a_k^{(i)}|s_k^{(i)}, g^{(i)})}} \gamma^t \frac{\nabla_\theta \pi_\theta(a_k^{(i)}|s_k^{(i)}, g')}{\pi_{\tilde{\theta}}(a_k^{(i)}|s_k^{(i)}, g')} A_{\tilde{\theta}}(a_k^{(i)}, s_k^{(i)}, g') \right] \tag{55}$$

$$s.t. \frac{1}{\lambda} \sum_{g' \in \mathcal{G}} \sum_{i=1}^N \sum_{t=0}^\infty \left[ \frac{\gamma^t}{2} \frac{\prod_{k=1}^t \frac{\pi_{\tilde{\theta}}(a_k^{(i)}|s_k^{(i)}, g')}{\pi_{\tilde{\theta}}(a_k^{(i)}|s_k^{(i)}, g^{(i)})}}{\sum_{j=1}^N \prod_{k=1}^t \frac{\pi_{\tilde{\theta}}(a_k^{(i)}|s_k^{(i)}, g')}{\pi_{\tilde{\theta}}(a_k^{(i)}|s_k^{(i)}, g^{(i)})}} (\log \pi_{\tilde{\theta}}(a_k^{(i)}|s_k^{(i)}, g') - \log \pi_\theta(a_k^{(i)}|s_k^{(i)}, g'))^2 \right] \leq \epsilon'. \tag{56}$$

We provide an explicit solution method for the optimization problem above in Appendix C.2.

While introducing weighted importance sampling may cause a certain level of bias which is identical to that of HPG (Rauber et al., 2019), the bias is to decrease in inverse ratio with regard to the increase of the data theoretically (Powell & Swann, 1966). Given the limited resources, we need to tradeoff between reducing bias and enlarging batch size. By picking a appropriate batch size, the improvement of weighted importance sampling is well demonstrated in the experiments.

### C.2  SOLUTION METHOD FOR HTRPO

For the HTRPO optimization problem, briefly denote the optimization problem in expression 55 and 56 as:

$$\max_\theta f(\theta)$$

$$s.t. \ g(\theta) \le \epsilon'. \tag{57}$$

For any policy parameter $\theta$ in the neighborhood of the parameter $\tilde{\theta}$, approximate the optimization problem with linear objective function and quadratic constraint:

$$\max_{\theta} \ f(\tilde{\theta}) + \nabla_{\theta} f(\tilde{\theta})(\theta - \tilde{\theta})$$

$$s.t. \ g(\tilde{\theta}) + \nabla_{\theta} g(\tilde{\theta})(\theta - \tilde{\theta}) + \frac{1}{2}(\theta - \tilde{\theta})^T \nabla_{\theta}^2 g(\tilde{\theta})(\theta - \tilde{\theta}) \le \epsilon'. \tag{58}$$

Noticeably, $g(\tilde{\theta}) = 0$ and $\nabla_{\theta} g(\tilde{\theta}) = 0$, which further simplifies the optimization problem to the following form:

$$\max_{\theta} \ f(\tilde{\theta}) + \nabla_{\theta} f(\tilde{\theta})(\theta - \tilde{\theta})$$

$$s.t. \ \frac{1}{2}(\theta - \tilde{\theta})^T \nabla_{\theta}^2 g(\tilde{\theta})(\theta - \tilde{\theta}) \le \epsilon'. \tag{59}$$

Given a convex optimzation problem with a linear objective function under a quadratic constraint, many well-practiced approaches can be taken to solve the problem analytically, among which we adopt the Karush-Kuhn-Tucker(KKT) conditions (Boyd et al., 2004). For a Lagrangian multiplier $\lambda$,

$$\frac{1}{2}(\theta - \tilde{\theta})^T \nabla_{\theta}^2 g(\tilde{\theta})(\theta - \tilde{\theta}) - \epsilon' \le 0$$

$$\lambda \ge 0$$

$$\lambda[\frac{1}{2}(\theta - \tilde{\theta})^T \nabla_{\theta}^2 g(\tilde{\theta})(\theta - \tilde{\theta}) - \epsilon'] = 0$$

$$-\nabla_{\theta} f(\tilde{\theta}) + \lambda \nabla_{\theta}^2 g(\tilde{\theta})(\theta - \tilde{\theta}) = 0 \tag{60}$$

Expressions in 60 form the KKT conditions of the optimization problem. Solving the KKT conditions,

$$\theta = \tilde{\theta} + \sqrt{\frac{2\epsilon'}{\nabla_{\theta} f(\tilde{\theta})^T [\nabla_{\theta}^2 g(\tilde{\theta})]^{-1} \nabla_{\theta} f(\tilde{\theta})}} [\nabla_{\theta}^2 g(\tilde{\theta})]^{-1} \nabla_{\theta} f(\tilde{\theta}) \tag{61}$$

The policies, however, in this paper are in the form of a neural network, which makes it extremely time-comsuming to compute the Hessian matrix. Thus, we compute $[\nabla_{\theta}^2 g(\tilde{\theta})]^{-1} \nabla_{\theta} f(\tilde{\theta})$ with conjugate gradient algorithm by solving the following equation:

$$[\nabla_{\theta}^2 g(\tilde{\theta})]x = \nabla_{\theta} f(\tilde{\theta}), \tag{62}$$

in which $[\nabla_{\theta}^2 g(\tilde{\theta})]x$ can be practically calculated through the following expansion:

$$[\nabla_{\theta}^2 g(\tilde{\theta})]x = \nabla_{\theta}[(\nabla_{\theta} g(\tilde{\theta}))^T x]. \tag{63}$$

# D  ALGORITHM

---

**Algorithm 1** Hindsight Trust Region Policy Optimization

---

**Input:**
    Training batch size $batchsize$
    Max number of training time steps $T_{max}$
    Policy $\theta$
    Q-function $\phi$
    Data Buffer $B_{origin}$ with its max size equal to $batchsize$           $\triangleright$ Initialization
**Output:**
    Optimized Policy $\theta^*$
 1: **for** $iteration = 1$ to $T_{max}/batchsize$ **do**
 2:     **while** $B_{origin}$ is not full **do**
 3:         Sample a trajectory $\tau = \{(s_t, a_t, r_t, s_{t+1}, g, \pi_\theta(a_t|s_t, g))\}_{t=1}^T$ using current policy $\theta$;
 4:         $B_{origin} = B_{origin} \cap \tau$;
 5:     **end while**                                    $\triangleright$ Collecting data
 6:     Sample alternative goals $\mathcal{G} = \{g'^{(i)}\}_{i=1}^{N_g}$ from achieved goals in $B_{origin}$;
 7:     $B_{train} = \varnothing$;
 8:     **for** $g'^{(i)}$ in $\mathcal{G}$ **do**
 9:         **for** $\tau$ in $B_{origin}$ **do**
10:             **for** $t = 0$ to T **do**
11:                 Compute $\pi_\theta(a_t|s_t, g'^{(i)})$;
12:                 Modify reward $r_t|g \rightarrow r_t|g'^{(i)}$;
13:             **end for**
14:             $\tau|g'^{(i)} = \{(s_t, a_t, r_t|g'^{(i)}, s_{t+1}, g, \pi_\theta(a_t|s_t, g), \pi_\theta(a_t|s_t, g'^{(i)}))\}_{t=1}^T$;
15:             $B_{train} = B_{train} \cap \tau|g'^{(i)}$;
16:         **end for**
17:     **end for**                              $\triangleright$ Generating training data
18:     Use $B_{train}$ to optimize policy $\theta$ with objective 55 and constraint 56 following Section C.2;
19:     $B_{origin} = \varnothing$;
20: **end for**
21: **return** optimized policy $\theta^*$;

---

# E   HYPERPARAMETERS

## E.1   HYPERPARAMETERS OF DISCRETE ENVIRONMENTS

Table 1: Hyperparameters of Discrete Environments

|                              | FlipBit8        | FlipBit16       | EmptyMaze       |
|------------------------------|-----------------|-----------------|-----------------|
| training time steps          | $5 \times 10^4$ | $5 \times 10^4$ | $5 \times 10^4$ |
| batch size                   | 128             | 256             | 256             |
| cg damping                   | 1e-2            | 1e-2            | 1e-3            |
| reward decay                 | 0.9             | 0.9             | 0.95            |
| max KL step                  | 1e-3            | 1e-3            | 1e-3            |
| critic optimizer             | Adam            | Adam            | Adam            |
| critic learning rate         | 5e-4            | 5e-4            | 5e-4            |
| critic updates per iteration | 10              | 10              | 10              |
| sampled goal number          | $\infty$        | $\infty$        | $\infty$        |

|                              | FourRoom        | FetchReach      | FetchPush       |
|------------------------------|-----------------|-----------------|-----------------|
| training time steps          | $5 \times 10^4$ | $5 \times 10^5$ | $2 \times 10^6$ |
| batch size                   | 256             | 800             | 800             |
| cg damping                   | 1e-3            | 1e-3            | 1e-3            |
| reward decay                 | 0.95            | 0.98            | 0.98            |
| max KL step                  | 1e-3            | 3e-6            | 3e-6            |
| critic optimizer             | Adam            | Adam            | Adam            |
| critic learning rate         | 5e-4            | 1e-4            | 1e-4            |
| critic updates per iteration | 10              | 10              | 10              |
| sampled goal number          | $\infty$        | 30              | 30              |

## E.2   HYPERPARAMETERS OF CONTINUOUS ENVIRONMENTS

Table 2: Hyperparameters of Continuous Environments

|                              | FetchReach      | FetchPush       | FetchSlide      |
|------------------------------|-----------------|-----------------|-----------------|
| training time steps          | $5 \times 10^5$ | $2 \times 10^6$ | $2 \times 10^6$ |
| batch size                   | 800             | 1600            | 3200            |
| cg damping                   | 1e-3            | 1e-3            | 1e-3            |
| reward decay                 | 0.98            | 0.98            | 0.98            |
| max KL step                  | 1e-5            | 1e-5            | 1e-5            |
| entropy weight               | 0               | 1e-4            | 1e-4            |
| critic optimizer             | Adam            | Adam            | Adam            |
| critic learning rate         | 5e-4            | 5e-4            | 5e-4            |
| critic updates per iteration | 20              | 20              | 20              |
| sampled goal number          | 100             | 100             | 100             |

# F    EXPERIMENTS

In this section, we provide a more comprehensive demonstration for the experiments of HTRPO. In detail, section F.1 narrates a full introduction to each environment; section F.2 gives out the sensitivity analysis of HTRPO including the performance under different network architectures and different numbers of alternative goals, in which we strictly adopt the control variable method and only the studied parameter is altered; section F.3 shows the supplementary materials of the experimental data including learning curves and success rates during the training process. We finetune the hyperparameters according to experience without hyperparameter search due to limited computing resources.

## F.1    ENVIRONMENTS

**k-Bit Flipping.** In each episode of this experiment, two arrays of length $k$ are generated. The first array is initialized with all 0's while the second one, usually regarded as the target array, is generated randomly. At each time step, the agent is able to flip one bit of the first array from 0 to 1 or from 1 to 0. Once the first array is exactly the same with the target array, the agent reaches the goal state and is then rewarded. The maximum number of time steps is $k$. In this experiment, we observe the performance of HTRPO under conditions that $k = 8$ and that $k = 16$ respectively. The general process of an 8-Bit Flipping task is demonstrated in Figure 1 (a).

**Grid World.** In this experiment, the agent starts at a position of an $11 \times 11$ grid with intransitable obstacles, and is trying to reach another randomly chosen position in this grid. The agent is allowed to move up, down, left and right at each time step. Moving into obstacles makes the agent remain in its current position. States of this environment is represented by 2-dimensional integral coordinates and the maximum number of time steps is 32. In *Empty Maze* environment, there is no impassable obstacles other than the outer boundary, and the agent starts at the upper-left corner of the grid. In *Four Rooms* environment (Sutton et al., 1999), walls separate the grid into 4 rooms, each with access to its adjacent rooms through single openings. Example cases of *Empty Maze* and *Four Rooms* environments adopted in this paper are demonstrated in Figure 1 (b) and (c).

**Fetch.** Fetch environment contains a 7-DoF Fetch robotic arm with a two-fingered parallel gripper in simulation(Plappert et al., 2018). In *Fetch Reach* environment, a target position is randomly chosen and the gripper of Fetch robotic arm needs to be moved upon it. In *Fetch Push*, the task for the robotic arm is to push a randomly placed block towards the goal state, anther randomly picked position, which is represented by a 3-dimensional Cartesian coordinate. In *Fetch Slide*, the robotic arm needs to exert a force on the block for it to slide towrds a chosen goal at a certain distance. A pictorial demonstration of this environment is shown in Figure 1 (d), in which the red dot represents the goal position. For the discrete Fetch environment, detailed settings follow that in Rauber et al. (2019); for the continuous version, the configurations of legal actions and states follow that in Plappert et al. (2018). The maximum number of time steps is 50. As in Plappert et al. (2018), we endure a 5cm target scope centered around the goal position for *Fetch Reach* and *Fetch Push*, and the tolerance is set as 20cm for *Fetch Slide* as in Andrychowicz et al. (2017).

## F.2 SENSITIVITY ANALYSIS

### F.2.1 DIFFERENT NETWORK ARCHITECTURES

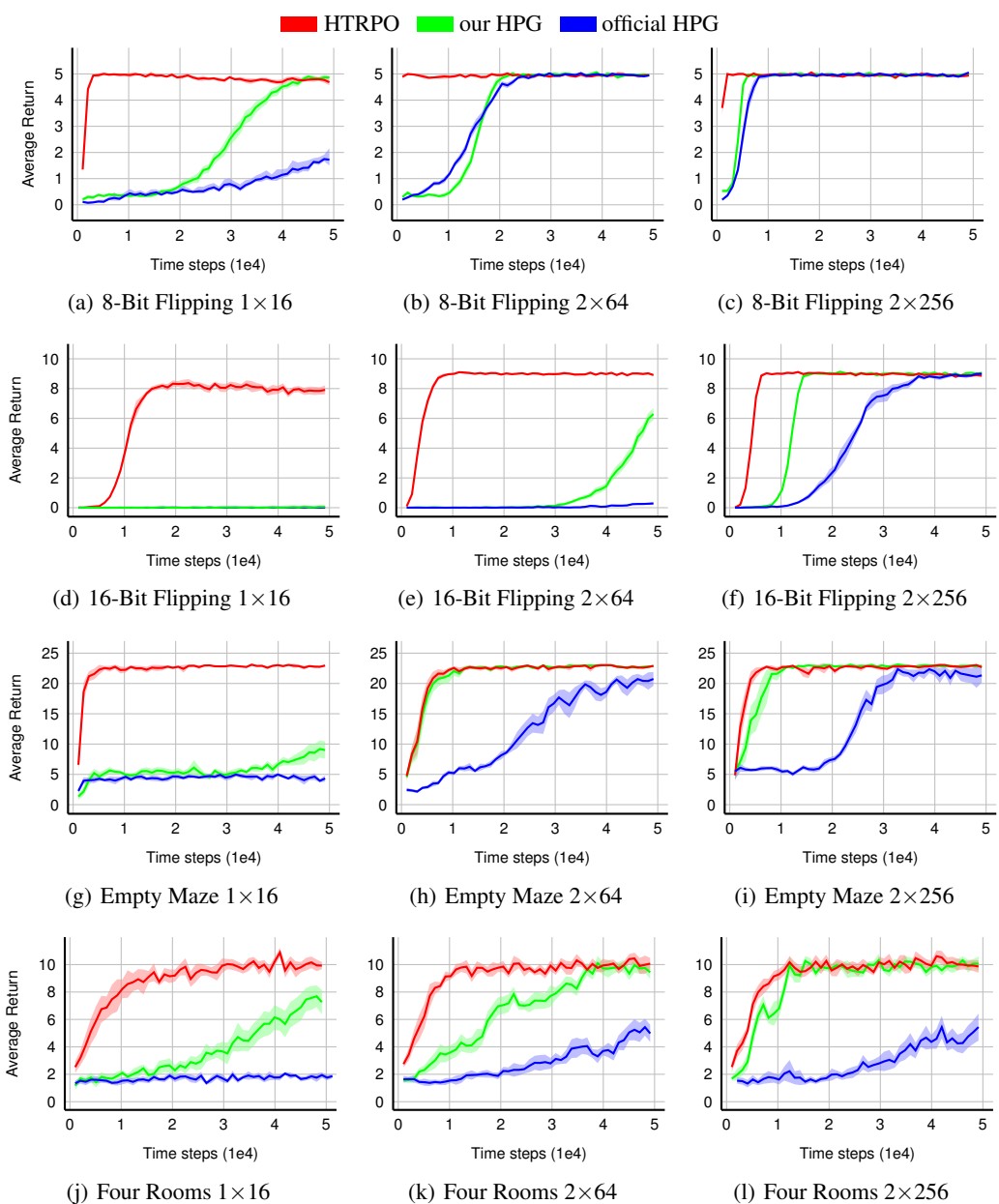

Figure 4: Evaluation curves with other network structures. Horizontally, 3 figures in each line illustrate the perfomances in one environment with different network architectures. Vertically, each column illustrate the performances of one kind of network architecture in different environments.

In this experiment, we observe the performance of HTRPO with different network architectures. Specially, we implement the proposed algorithm under 3 different network settings, i.e. networks with a 16-unit layer, two 64-unit layers and two 256-unit layers respectively. For the record, all parameters and other settings remain the same aside from the network architecture. As demonstrated in Figure 4, each row shows the performance under different network architecture settings for each environment. A general conclusion can be drawn that networks with more hidden layers and more neurons help to speed up the convergence. However, one difference is that for HTRPO, it converges

quickly in all the settings, while for HPG, it converges much slower especially when the network architecture is simple. We believe that the iteratively searching of optimal solution in the trust region helps the network converge rapidly and is more robust to different network architecture.

### F.2.2 DIFFERENT NUMBER OF ALTERNATIVE GOALS

In this experiment, how the number of alternative goals, as a key parameter, affects the performance of HTRPO is studied. We conduct all the experiments, both discrete and continuous with different number of alternative goals. For discrete environments, we set the number of alternative goals to be 10, 30, 100 and $\infty$ in turn. For continuous environments, we compare the performance under 10, 30, 100 alternative goals respectively. The evaluation curves are shown in 5. From the results, we can see that in simple discrete environments, $\infty$ alternative goals produce the fastest convergence. In complex and continuous environments, 30 and 100 alternative goals lead to comparatively good performance. It is not hard to see that Hindishgt TRPO with more alternative goals achieves better converging speed, which may be credited to the corresponding increase on training samples. This is, to some extent, similar to data augmentation.

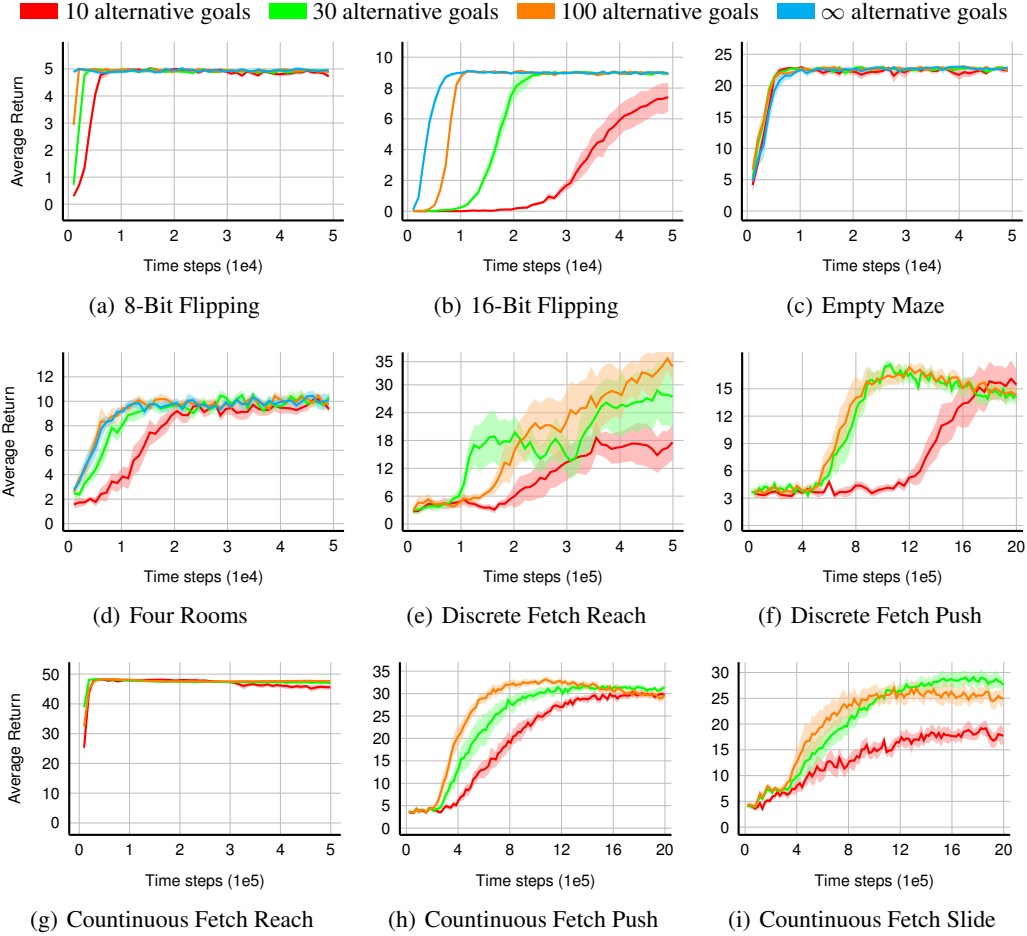

Figure 5: Evaluation curves for different number of alternative goals: 8-Bit Flipping, 16-Bit Flipping, Empty Maze, Four Rooms, Discrete Fetch Reach, Discrete Fetch Push, Continuous Fetch Reach, Continuous Fetch Push and Contiuous Fetch Slide. The full lines represent the average evaluation over 10 trails and the shaded regions represent the corresponding standard deviation.

## F.3 COMPREHENSIVE TRAINING CURVES

### F.3.1 TRAINING CURVES

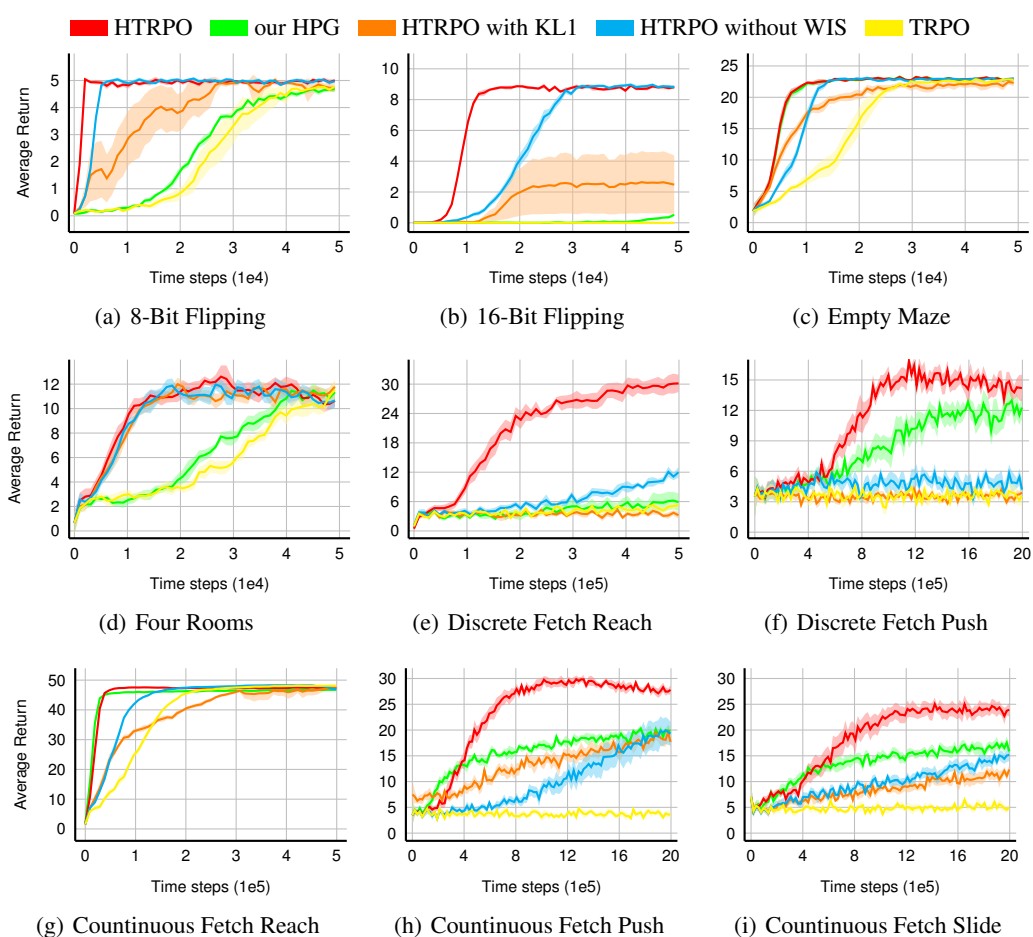

Figure 6: Training curves for all environments: 8-Bit Flipping, 16-Bit Flipping, Empty Maze, Four Rooms, Discrete Fetch Reach, Discrete Fetch Push, Continuous Fetch Reach, Continuous Fetch Push and Contiuous Fetch Slide. The full lines represent the average evaluation over 10 trails and the shaded regions represent the corresponding standard deviation.

### F.3.2 SUCCESS RATE

In this section, we demonstrate the success rates of HTRPO during both evaluation and training. For the record, the actions during the training process are sampled by the distribution output by the network while during the evaluation process, we adopt a greedy strategy to choose the action by the mean value of the distribution. Table 3 lists the success rates of Fetch Push and Fetch Slide during evaluation, in which the ultimate values reflect the mean computed with 1000 test results in each iteration. They are the only two environments listed for they are the most complex ones. Figure 7 illustrates the success rate curves during the training process.

Table 3: Evaluation success rate for Fetch Push and Fetch Slide

| Time step | Fetch Push | | Fetch Slide | |
|---|---|---|---|---|
| | our HPG (%) | HTRPO (%) | our HPG (%) | HTRPO (%) |
| 480k | 56.4 | **63.2** | 36.1 | **46.4** |
| 960k | 65.6 | **91.9** | 59.5 | **79.9** |
| 1920k | 87.2 | **89.7** | 61.2 | **82.5** |

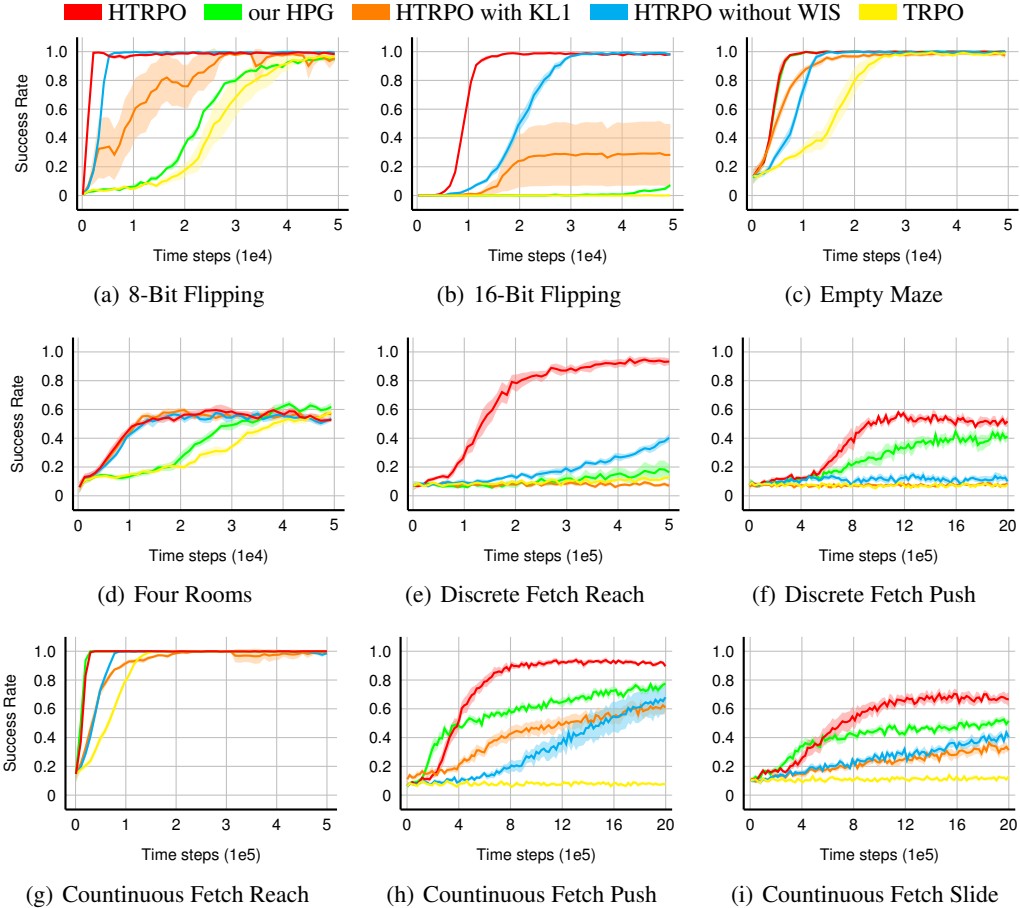

Figure 7: Success Rate for all environments:8-Bit Flipping, 16-Bit Flipping, Empty Maze, Four Rooms, Discrete Fetch Reach, Discrete Fetch Push, Continuous Fetch Reach, Continuous Fetch Push and Contiuous Fetch Slide. The full lines represent the average evaluation over 10 trails and the shaded regions represent the corresponding standard deviation.

## F.4    COMPARISON WITH DENSE REWARD TRPO

Figure 8 demonstrates the success rate for HTRPO, TRPO with sparse reward and TRPO with dense reward for Continuous Fetch Reach, Continuous Fetch Push and Contiuous Fetch Slide. The performance of TRPO with dense rewards is similar to that of TRPO with sparse rewards: for FetchReach, it converges much more slowly than HTRPO while for FetchPush and FetchSlide, it doesnt work in the whole training process (2 million time steps). Similar conclusions can also be found in some other literatures of Hindsight (Plappert et al., 2018). Therefore, it can be concluded that HTRPO outperforms it significantly.

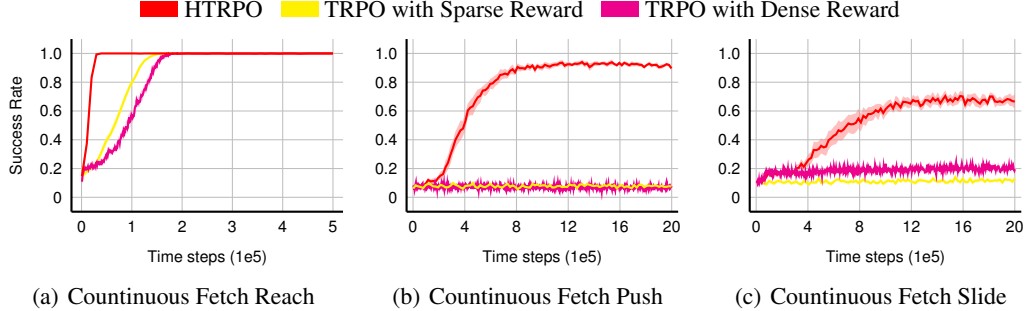

(a) Countinuous Fetch Reach          (b) Countinuous Fetch Push          (c) Countinuous Fetch Slide

Figure 8: Success Rate for HTRPO, TRPO with Sparse Reward and TRPO with Dense Reward for Continuous Fetch Reach, Continuous Fetch Push and Contiuous Fetch Slide. The full lines represent the average evaluation over 10 trails and the shaded regions represent the corresponding standard deviation.

### F.5 COMPARISON OF DIFFERENT KL EXPECTATION ESTIMATORS

Figure 9 demonstrates the estimation results of KL divergence expectation in the training process. From these data we can see that in the experiments, the approximation of equation 13 can significantly reduce the variance of KL expectation estimation. Besides, the comparison of performance between HTRPO and HTRPO with KL1 also shows the efficiency of this approximation, which helps improve the final performance significantly. Both "HTRPO" and "HTRPO without WIS" use the estimation method in equation 13 with one difference being that "HTRPO without WIS" doesn't adopt weighted importance sampling. Thus, from Figure 9, we can see that "HTRPO" demonstrates the least variance.

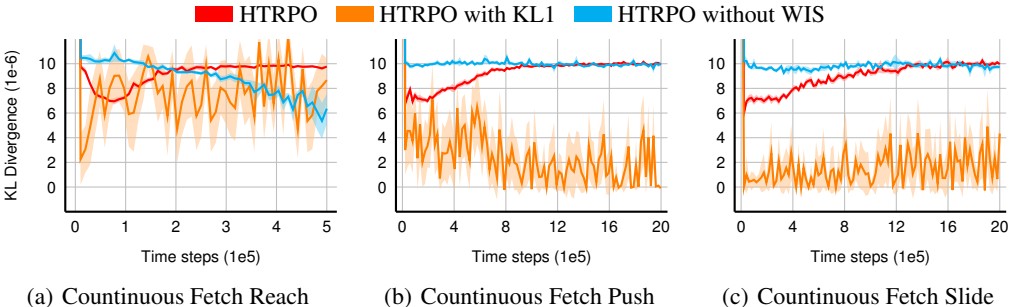

    (a) Countinuous Fetch Reach     (b) Countinuous Fetch Push     (c) Countinuous Fetch Slide

Figure 9: Estimation of KL Divergence Expectation for different variants of HTRPO in Continuous Fetch Reach, Continuous Fetch Push and Contiuous Fetch Slide. The full lines represent the average evaluation over 10 trails and the shaded regions represent the corresponding standard deviation. The curves for KL1 are comparatively lower than those of equation 13. Note that in TRPO, the linear search mechanism adjust the updating step size according to the estimation of KL divergence expectation. It sets a threshold to constrain the KL divergence. For those above the threshold, the updating step size will be reduced to ensure that the estimation of KL divergence estimation falls within the threshold. This explains why the curves for KL1 are comparatively lower. However, since the estimation of KL divergence expectation in HTRPO falls near the expected value, such step size adjustment is rarely triggered. This benefits from the much lower variance of equation 13.

