# OpenReview forum: "Hindsight Trust Region Policy Optimization"
_ICLR.cc/2020/Conference — Reject_

### Official Review · AnonReviewer1 · 2019-10-14
**Official Blind Review #1**

**Rating:** 3

**Review:**

The paper proposes an extension to TRPO that makes use of hindsight experience replay.  The technique follows previous work on HER for policy gradient methods, augmenting these previous approaches with a constraint based on the average squared difference of log-probabilities. Experiments show that the method can provide significant improvements over previous work.

Overall, the paper provides a convincing demonstration of trust region principles to goal-conditioned HER learning, although I think the paper could be improved in the following ways:
-- Bounding KL by the squared difference of log-probabilities seems loose.  The original TRPO objective is based on a TV-divergence (and before that, based on a state-occupancy divergence). Is it possible to directly bound the TV-divergence (either of actions or state occupancies) by a squared difference of log-probabilities?
-- The use of WIS greatly biases the objective. Is there a way to quantify or motivate this bias?
-- What is the method in which the alternative goal g' is proposed? I believe this can have a great effect on the bias and variance of the proposed method.

**Experience Assessment:**

I have published one or two papers in this area.

**Review Assessment: Checking Correctness Of Derivations And Theory:**

I assessed the sensibility of the derivations and theory.

**Review Assessment: Checking Correctness Of Experiments:**

I assessed the sensibility of the experiments.

**Review Assessment: Thoroughness In Paper Reading:**

I read the paper at least twice and used my best judgement in assessing the paper.

---

> ### Author Response · Authors · 2019-11-15
> **Re: Official Blind Review #1**
>
> Thanks for your review.
>
> Q1 Bounding KL by the squared difference of log-probabilities seems loose. The original TRPO objective is based on a TV-divergence (and before that, based on a state-occupancy divergence). Is it possible to directly bound the TV-divergence (either of actions or state occupancies) by a squared difference of log-probabilities?
>
> This is quite an enlightening idea and an alternative way to develop a variant of H-TRPO. In fact, it is possible to directly bound the square of TV-divergence and further improve the performance of H-TRPO. However, there may be some problems. For example, TV-divergence is defined as $E[|p-q|]$, which is in the form of the expectation of an absolute value. It is not differentiable at point zero, which may lead to numerical instability in calculating the derivative of TV divergence w. r. t. parameter $\theta$ when the two distributions are close. As you suggested, there may be a solution to this problem, for example, by approximating TV-divergence with other forms, which however needs a whole different solving process. As promising as it may seem to regard it as a future work, it is sufficiently proved, both in theories and experiments including the discrete ones HPG is tested on and continuous ones like FetchPush and FetchSlide, that our proposed algorithm is effective. Finally, returning to your question, we agree that it is possible to bound the TV-divergence directly.
>
> Q2 The use of WIS greatly biases the objective. Is there a way to quantify or motivate this bias?
>
> Thanks for your question. We add some discussion on the motivation of this bias in Appendix C.1. As explained in the Appendix, bias is to decrease in inverse ratio with regard to the increase of the data. Given the limited resources, we need to tradeoff between reducing bias and enlarging batch size. Doing the best we can, there exists a certain level of bias, which is identical to that of HPG (Rauber et.al. 2019). However, the improvement of WIS is well demonstrated in the experiments.
>
> Q3 What is the method in which the alternative goal g' is proposed? I believe this can have a great effect on the bias and variance of the proposed method.
>
> Thanks for your question. We follow the alternative goal sampling method of HPG (Rauber et.al. 2019), which subsamples a set of alternative goals in all achieved states. As explained in HPG:
> “Secondly, although it is feasible to compute Expression 12 exactly when the goal distribution is known, we sometimes subsample the sets of active goals per episode.”
> We add relevant clarification in Appendix C.1:
> “In experiments, we follow the alternative goal sampling method of HPG.”
>
> As for its effect on variance and bias,
> 1. For variance, the lager the hindsight data size is, the lower the variance will be no matter for the objective or the KL expectation constraint. This statement is also verified in the experiments demonstrated in Appendix F.2.2. As a result, we choose a sufficiently large number of alternative goal which leads to good performance. For the record, we set the same number of alternative goals for different algorithms in the comparisons for fair.
>
> 2. For bias, thinking from the perspective of the distribution of the original goals, this kind of alternative goal sampling method introduces some bias. Yet, from the perspective of the distribution of the alternative goals, such bias no longer exists. In other words, if the goals of training data follow the alternative distribution, the objective and KL expectation will be estimated w.r.t. the alternative goal distribution. Therefore, during the learning process, our algorithm is encouraging the agent to achieve the alternative goals. Such a mechanism serves as a mutual approach for all hindsight methods [1-3], which can be seen as a merit, for the intention is to guide the agent to achieve the alternative goals and then generalize to the original goals. We also add the relevant description in the final paragraph of Appendix C.1.
>
> [1]Matthias Plappert, et al. Multi-goal reinforcement learning: Challenging robotics environments and request for research. arXiv preprint arXiv:1802.09464, 2018.
> [2]Paulo Rauber, et al. Hindsight policy gradients. In International Conference on Learning Representations, 2019.
> [3]Marcin Andrychowicz, et al. Hindsight experience replay. In Advances in Neural Information Processing Systems, pp. 5048–5058, 2017.

---

### Official Review · AnonReviewer2 · 2019-10-16
**Official Blind Review #2**

**Rating:** 3

**Review:**

*Summary* : This paper augments the TRPO policy optimization objective with hindsight data, where the hindsight data is generated from goals based on trajectories. The key contribution of the paper is based on deriving an on-policy adaptation of hindsight based TRPO, that can be useful for sparse reward environments. The paper draws ideas from existing papers such as HPG and considers the IS based variant of HPG for on-policy, similar to TRPO, that can achieve monotonic performance improvements. Furthermore, the authors introduce a logarithmic form of constraint, by re-deriving the KL constraint and leading to a f-divergence based constraint, which is argued to have useful effects in terms of lowering the variance. Experimental results are compared with baselines including HPG and its variants on standard sparse reward benchmark tasks.

*Comments* :

	- The core contribution of the paper is to introduce hindsight based TRPO where end states from the trajectory treated as goals can be useful for generating pseudo-rewards, such that existing TRPO based approaches can be better suited for sparse reward environments. The claim is that existing policy gradient methods cannot sufficiently explore in sparse reward environments.
	- Since incorporating hindsight data can make the approach off-policy in nature, leading to higher variance and instability, the authors propose to augment hindsight approach into on-policy based methods such as TRPO. The key contribution is to develop a theoretically sound approach for hindsight based policy optimization.
	- The hindsight TRPO objective is derived based on goal-conditioned policies with TRPO, based on existing recent work on Hindsight Policy Gradient (HPG) (Rauber et al., 2019). The key difficulty that needs to be tackled is when introducing hindsight experience makes the approach off-policy in nature. The authors derive the HTRPO objective based on an importance sampling approach that can also incorporate the hindsight data.
	- Equation 7 writes out the key quantity, an IS based approach considering the current goal and alternative goals to derive a similar TRPO objective based on IS based Monte-Carlo estimators, while maintaining the trust region guarantees. Equation 8 further shows the gradient of the objective. Theorem 3.1 and 3.2 follows naturally where the key trick in Thm 3.1 is going from equation 19 to 20 to derive the IS based estimator with goal conditioned trajectories. I am not sure why Thm 3.2 needs to be written out explicitly, given that it follows naturally for gradient of the expectation? Is there any key takeaway from it?
	- In TRPO, the expectation is based on states sampled from a state distribution. The authors argue that for hindsight based data, this state distribution in fact can change due to changes in the generated goals (ending states), and hence the KL needs to be in expectation w.r.t to the state occupancy measure. Furthermore, the authors change the KL based constraint into a logarithmic form of a constraint such as to reduce variance and introduce more stability. To achieve this, the paper uses an approximation to the logarithmic form of the constraint, by using an expectation of the square instead of plain differences between the log of policies. The key is that instead of using the KL divergence, the authors introduce f-divergence where the function is convex allowing smooth optimization.
	- The overall contribution of the paper can be summarized from equation 15 - introducing a IS based correction, while remaining on-policy for hindsight based TRPO objective. And since hindsight can change the underlying state distribution, leading to more instability, the paper introduces a different form of constraint (based on the f-divergence) which can have lower variance than the KL form of constraint.
	- Experimental results are demonstrated for few sparse reward benchmarks, comparing to the standard HPG, TRPO and several variants of the proposed HTRPO approach with weighted IS and with the KL constraint. The advantages of HTRPO on these tasks seems clear, mainly in the Fetch Reach and Fetch Push tasks, significantly outperforming the baselines. Even in the continuous tasks, HTRPO seems to outperform the baselines consistently.

*Feedback/Questions* :

	- I am not sure of the significance of Theorem 3.2 - it seems obvious that the gradient of the objective spans out naturally from equation 7?
	- The authors mention about the state occupancy measure instead of considering the state distribution for the KL term. However, the discussion of state occupancy measure seemed to have faded away? What was the significance of mentioning it, or why should it be considered even? There are no assumptions being made on whether the state distribution should be the discounted state occupancy measure or the stationary distribution (if infinite horizon is considered).
	- The introduction of the logarithmic form of constraint, even though shows theoretically to reduce variance, is not well motivated or demonstrated from the suite of experimental results? From the results, it is not obvious whether this form of constraint is indeed having useful effects in terms of reduced variance?
	- The paper seems to adapt from the HPG objective, and does indeed a great job comparing to HPG throughout the suite of experiments. However, in the results, the paper mainly compares to other off-policy based methods including HPG and its variants (official and re-implemented one). I find the comparison of results a bit odd in that sense, since it is comparing the on-policy adaptation of HPG (ie, HTRPO) and the off-policy variants? If run for long enough, does all converge to the same results? If so, then the benefits is mainly in faster learning (assumably due to better exploration in the sparse reward tasks). But then again, these benefits may be because of the on-policy approach compared to the off-policy one?
	- I would encourage the authors to compare to more standard goal-conditioned or reward shaping based baselines for TRPO. For example, does the proposed HTRPO approach perform better compared to other goal-conditioned approaches of TRPO, or for example if a form of reward shaping (based on goals) are used in TRPO? It would then be a more likewise comparison? The current results seem to show benefits of HTRPO, but I think there is a need for stronger baselines where TRPO + exploraton (reward shaping or goal conditioning) performs better?
	- I am not convinced about the arguments with sensitivity of HTRPO with network architecture and parameters. How is this demonstrated from the suite of results?

*Summary of Review and Score* :

Overall, I think the paper has merits, mainly in terms of deriving the on-policy adaptation with hindsight data. The key objectives are derived nicely in the write-up and easy to follow, although there are some confusions that need to be clarified (example : the discussion on state occupancy measure and the significance of it). The paper motivates exploration for TRPO in sparse reward tasks, and considers the hindsight adaptation of existing TRPO. But related works such as HPG have already taken a similar approach for the off-policy case, and this paper's key contribution is in terms of theoretically deriving the objectives for on-policy adaptation. However, I am not convinced about the overall merits and contributions of the paper, especially in terms of demonstrated results and proper comparisons with baselines. I think while the objectives and derivations follow naturally, the contributions of the paper is somewhat marginal.

I would therefore vote to marginally reject this paper - mainly in light of the core novel contribution and due to lack of sufficient results demonstrating the usefulness of the approach. The paper combines several things together - especially discussions of the logarithmic form of the constraint. I doubt whether all these introduced together led to the improvements shown in the experimental results or not. It would be useful to clarify the contributions from each of the components.



**Experience Assessment:**

I have published one or two papers in this area.

**Review Assessment: Checking Correctness Of Derivations And Theory:**

I carefully checked the derivations and theory.

**Review Assessment: Checking Correctness Of Experiments:**

I carefully checked the experiments.

**Review Assessment: Thoroughness In Paper Reading:**

I read the paper thoroughly.

---

> ### Author Response · Authors · 2019-11-15
> **Re: Official Blind Review #2 part(1/2)**
>
> Thanks for your detailed advice.
>
> Q1 I am not sure of the significance of Theorem 3.2 - it seems obvious that the gradient of the objective spans out naturally from equation 7?
>
> Thanks for your kind question. In fact, we want to give the readers some insights of the context of the whole paper only through all the theorems. Though Theorem 3.2 is straightforward from Eqn. 7, it demonstrates the gradient of the objective directly, which plays an important role in the policy optimization procedure.
>
> Q2 What was the significance of mentioning state occupancy measure, or why should it be considered even?
>
> Thank you for your question. Though the relevant definitions are in the first paragraph of Section 2, there may indeed be some confusion. To further address the significance of “state occupancy measure” and clear some misunderstandings:
>
> 1. Assume the “state occupancy measure” in your first question refers to $\gamma$-discounted state visitation distribution in the paper, which was already been introduced in TRPO. As we know, in TRPO, the maximum of KL-divergence is approximated by the expectation of KL-divergence over $\gamma$-discounted state visitation distribution, which is represented by $\rho(s)$ in this paper.
>
> 2. Another possibility is that by “state occupancy measure”, you mean “occupancy measure”, which is represented by $\rho(s,a)$. Throughout the paper, it is our negligence that we didn’t use $\rho(s,a)$ to represent occupancy measure. However, “$s~\rho(s), a~\pi(a|s)$” is equivalent to $\rho(s,a)$. We have modified all relevant representations in the paper. As for how it is derived into this form instead of stationary distribution, we first expand the KL divergence using the log form, as shown in the proof of Theorem 4.1. After that, the action distribution under the condition of s can be naturally and straightly merged with the \gamma-discounted state visitation distribution and it will become the occupancy measure in theory. We do not assume anything else because we just follow the idea in TRPO using the expectation of KL to approximate maximum of KL, and derives all the remaining conclusions rigorously.
>
> Q3 The introduction of the logarithmic form of constraint is not well motivated or demonstrated from the suite of experimental results?
>
> Thanks for your question. We append the corresponding data in Appendix F.5. From these data, we can see that in the experiments, this approximation can significantly reduce the variance of KL expectation estimation. Besides, the comparison of performance between “HTRPO” and “HTRPO with KL1” also shows the efficiency of this approximation, which helps improve the final performance significantly. In fact, the corresponding experimental data for continuous tasks are preserved and we didn’t put them in the paper because we thought that the theoretical proof of variance reduction was enough. Any further doubts are welcomed and we will be responsible for every conclusion in this paper.

---

> > ### Author Response · Authors · 2019-11-15
> > **Re: Official Blind Review #2 part(2/2)**
> >
> > Q4. HTRPO is not well compared with HPG.  And if run for long enough, does all converge to the same results?
> >
> > Thanks for your questions.
> >
> > Q4.1. "I find the comparison of results a bit odd in that sense, since it is comparing the on-policy adaptation of HPG (ie, HTRPO) and the off-policy variants?"
> >
> > In fact, in either HTRPO or HPG, it is accordant that they both use off-policy data (e.g. hindsight data) to train on-policy algorithms. Therefore, they both adopt importance sampling as a necessary approach to correct the discrepancy between off-policy data and on-policy data. To argue the main difference between them, in HTRPO, the policy optimization procedure is implemented by iteratively searching the locally optimal solution for the objective while in HPG, the policy is just updated using policy gradients. Therefore, we think that the comparison is fair enough to show the advantages of HTRPO over HPG.
> >
> > Q4.2. "If run for long enough, does all converge to the same results?"
> >
> > The final performance of HTRPO is better. Admittedly, as claimed in the paper, in simple environments like Bit Flipping and Grid World, they converge to the same results, because these environments are too simple to show the advantage of HTRPO over HPG. However, in more complex environments like FetchPush and FetchSlide, it is obvious that HTRPO outperforms HPG significantly, both in convergence speed and final performance as shown in the experiments.
> >
> > Q5 I would encourage the authors to compare to more standard goal-conditioned or reward shaping based baselines for TRPO.
> >
> > Thanks for your question. HTRPO outperforms the goal-conditioned TRPO with reward reshaping significantly. We add the experiments in the Appendix. In fact, the TRPO in this paper is goal-conditioned TRPO, because all the tasks mentioned in this paper are goal-conditioned. And in Appendix C.4, we also add the results of goal-conditioned TRPO training with dense rewards. We test TRPO with dense rewards in FetchReach, FetchPush and FetchSlide environments. The performance is similar to that of TRPO with sparse rewards: for FetchReach, it converges much more slowly than HTRPO while for FetchPush and FetchSlide, it doesn’t work in the whole training time (2 million time steps). Similar conclusions can also be found in some other literatures of Hindsight (Plappert et al. 2018). Therefore, it can be concluded that HTRPO outperforms it significantly.
> >
> > [1] Matthias Plappert, et al. Multi-Goal Reinforcement Learning: Challenging Robotics Environments and Request for Research. Arxiv. 2018.
> >
> > Q6 I am not convinced about the arguments with sensitivity of HTRPO with network architecture and parameters. How is this demonstrated from the suite of results?
> >
> > Thanks for your question. In the experiments shown in Appendix F.2.1, each row shows the performance under different network architecture settings for each environment. A conclusion can be drawn that fully-connected policy networks with more hidden layers and more neurons help to speed up the convergence. However, one difference is that for HTRPO, it converges quickly in all the settings, while for HPG, it converges much slower especially when the network has fewer neurons (e.g. one hidden layer with 16 units). We believe that the iteratively searching of the optimal solution in the trust region helps the network converge rapidly and is less affected by different network architecture. The discussion above is added to Appendix F.2.1.

---

### Official Review · AnonReviewer3 · 2019-10-28
**Official Blind Review #3**

**Rating:** 6

**Review:**

Summary

The paper builds on top of prior work in hindsight policy gradients (Rauber et.al.) and trust region policy optimization (Schulman et.al.), proposing a hindsight trust region policy optimization. Conceptually this direction makes a lot of sense, since hindsight is in general shown to be useful when training goal conditioned policies. The formulation generally  appears to be sound and is a straightforward extension of the importance sampling techniques from Rabuer et.al. to the TRPO setting. Experimental results show that the proposed changes bring significant improvements over baselines on sparse reward settings.

Strengths:
+ The paper appears well formulated, and well motivated
+ The experimental results appear quite strong.
+ Description of the experimental details is quite clear.

Weaknesses:
- Most of the weaknesses that I can find are in terms of the presentation and writing which could be improved. Specifically, it would be good to clarify when writing the HTRPO equation (Eqn. 7) that this is still a constrained optimization problem with a KL divergence between polcies. Further, it should be better clarified and justified which KL divergence the constraint should be between, since there are two choices \pi(a| s, g) or \pi(a| s, g’).
- It would be make the results section flow much better if the paper were to adopt g as the original goal, and g’ as the alternative goal (which makes a much better flow from the non-hindsight case to the hindsight case).
- It seems a bit wasteful to mention Theorem 4.1 as a theorem, since it does not feel like a major result and is a straightforward monte carlo estimation of the KL divergence.
- Missing baseline: it would be nice to check if the method of estimating the KL divergence using difference of squares of log probs (Eqn. 12) improves TRPO (and a clarification on whether this is would be a novel application in the first place). This might be a result of independent interest outside of the hindsight context.


**Experience Assessment:**

I do not know much about this area.

**Review Assessment: Checking Correctness Of Derivations And Theory:**

I assessed the sensibility of the derivations and theory.

**Review Assessment: Checking Correctness Of Experiments:**

I assessed the sensibility of the experiments.

**Review Assessment: Thoroughness In Paper Reading:**

I read the paper at least twice and used my best judgement in assessing the paper.

---

> ### Author Response · Authors · 2019-11-15
> **Re: Official Blind Review #3**
>
> Thanks for the enlightening reviews.
>
> Q1. Writing Weakness.
>
> Thanks for your constructive advice. We have carefully proofread our paper and improved our writing. Specifically, for your two suggestions:
>
> Suggestion1:
> "it would be good to clarify when writing the HTRPO equation (Eqn. 7) that this is still a constrained optimization problem with a KL divergence between policies."
> We have added the clarification following Theorem 3.1:
> “It will be solved under a KL divergence expectation constraint, which will be discussed in detail in Section 4.”
> to emphasize that this is in fact an objective of a constrained problem.
>
> Suggestion2:
> "it should be better clarified and justified which KL divergence the constraint should be between, since there are two choices $\pi(a| s, g)$ or $\pi(a| s, g’)$."
> From my view, the constraint is an expectation of KL divergence between old and new policies over all possible goals and states. If you are asking about the goal condition in each KL divergence, the answer is that the condition is the alternative goal, i.e., $\pi(a| s, g’)$. We also add some corresponding description following Theorem 4.4.:
> “In the experiments, we use hindsight data under condition g' to estimate the expectation.”
>
> Q2 It would be make the results section flow much better if the paper were to adopt g as the original goal, and g’ as the alternative goal (which makes a much better flow from the non-hindsight case to the hindsight case).
>
> Thanks for your advice. As indicated in Theorem 4.4 of the paper, we indeed adopt g’ as the alternative goal. In detail, all the training trajectories are sampled using the original goal as the condition, and the distribution of the trajectories will change when we modify them to obtain hindsight data, which is conditioned on alternative goals. Formally, the distribution of sampling is represented by $p(\tau|g)$, and after modification, the actual distribution will be $p(\tau|g’)$. In fact, we want to compute $E_{g’}[E_{p(\tau|g’)}[D_{KL}]]$, which is the expectation of KL over all potential goals. However, we do not sample trajectories using $p(\tau|g’)$, therefore, we need to introduce the importance sampling to correct the discrepancy, which will result in $E_{g’}[E_{p(\tau|g)}[w * D_{KL}]]$ where w is the importance weight.
>
> Q3 It seems a bit wasteful to mention Theorem 4.1 as a theorem, since it does not feel like a major result and is a straightforward monte carlo estimation of the KL divergence.
>
> Thanks for your question. In fact, we want to highlight this straightforward conclusion because it is the foundation of all the later derivation, and it is the main difference of the estimation methods for KL divergence expectation between TRPO and HTRPO. We want to give the readers some insights of the context of the whole paper only through all the theorems.
>
> Q4 Missing baseline: it would be nice to check if the method of estimating the KL divergence using difference of squares of log probs (Eqn. 12) improves TRPO (and a clarification on whether this is would be a novel application in the first place). This might be a result of independent interest outside of the hindsight context.
>
> Thanks for your advice. It is definitely worthwhile to consider the baseline you proposed. As a matter of fact, the TRPO results demonstrated in the paper are indeed conducted with the constraint in the form of Eqn.12. We did not include TRPO with its original constraint, i.e. the left side of Eqn9, for that we find the results basically identical to that of Eqn.12 constraint. If needed, our open-source code implemented the switch for these two different constraints, to be specific, line 344-354 in package HTRPO agents/PG.py. Moreover, we have tested them on some other environments like Mujoco Hopper, and obtained similar results. For reasons above, we have fairly compared HTRPO with the baseline you proposed.
>
> To illustrate the contribution of estimating the KL divergence using the difference of squares of log probs (Eqn. 12), other than maintaining the high performance and low variance, there are actually two key merits compared with the left of Eqn.9. Firstly, starting from the defect of the left side of Eqn.9, the KL divergence needs to be analytically computed, which may not always be possible, for example, the KL divergence between GMMs. On the contrary, such an assumption is unnecessary for Eqn.12, for its origination from estimations. Secondly, when the training data are off-policy, the state distribution may inevitably change and the method from the left side of Eqn.9 for estimating the expectation of KL divergence is no longer valid. This discrepancy cannot be corrected. However, if using Eqn.12 to estimate the expectation of KL divergence, it can be corrected using importance sampling.

---

### Author Response · Authors · 2019-11-15
**Emphasis on the contributions**

We would like to emphasize the contributions of this paper.
1. We propose a methodology called Hindsight Trust Region Policy Optimization (HTRPO). In HTRPO, a hindsight form of policy optimization problem within trust region is theoretically derived, which can be approximately solved with the Monte Carlo estimator using severely off-policy hindsight experience data.
2. To overcome the high variance and instability in KL divergence estimation, another f -divergence is applied to approximate KL divergence, and both theoretically and practically, it is proved to be more efficient and stable. The new estimation method can have a wider usage maintaining the learning stability.
3. We first extend policy gradient methods with hindsight to continuous tasks such as FetchPush and FetchSlide and verified the effectiveness of HTRPO. It answers an open question in [1].

[1] Matthias Plappert, et al. Multi-goal reinforcement learning: Challenging robotics environments and request for research. arXiv preprint arXiv:1802.09464, 2018.

---

### Decision · Program_Chairs · 2019-12-19

**Decision:**

Reject

**Comment:**

The paper pursues an interesting approach, but requires additional maturation.  The experienced reviewers raise several concerns about the current version of the paper.  The significance of the contribution was questioned.  The paper missed key opportunities to evaluate and justify critical aspects of the proposed approach, via targeted ablation and baseline studies.  The quality and clarity of the technical exposition was also criticized.  The comments submitted by the reviewers should help the authors strengthen the paper.